# ON THE CRUCIAL ROLE OF INITIALIZATION FOR MATRIX FACTORIZATION

**Bingcong Li[1], Liang Zhang[1], Aryan Mokhtari[2], Niao He[1]**
[1]ETH Zurich, [2]The University of Texas at Austin
{bingcong.li,liang.zhang,niao.he}@inf.ethz.ch
mokhtari@austin.utexas.edu

## ABSTRACT

This work revisits the classical low-rank matrix factorization problem and unveils the critical role of initialization in shaping convergence rates for such nonconvex and nonsmooth optimization. We introduce Nyström initialization, which significantly improves the global convergence of Scaled Gradient Descent (ScaledGD) in both symmetric and asymmetric matrix factorization tasks. Specifically, we prove that ScaledGD with Nyström initialization achieves quadratic convergence in cases where only linear rates were previously known. Furthermore, we extend this initialization to low-rank adapters (LoRA) commonly used for finetuning foundation models. Our approach, NoRA, i.e., LoRA with Nyström initialization, demonstrates superior performance across various downstream tasks and model scales, from 1B to 7B parameters, in large language and diffusion models.

## 1 INTRODUCTION

Compared with learning rates and descent directions, initialization has been a relatively overlooked aspect of optimization. In the widely studied smooth optimization literature (Nesterov, 2004; Ghadimi & Lan, 2013), as long as a suitable (small) learning rate is chosen, most of optimization algorithms such as GD provably converge to a stationary point at the same rate, regardless of initialization. This work goes beyond stationary points and highlights the crucial role of initialization for global optimality of Burer-Monteiro factorization (Burer & Monteiro, 2003) – *the same algorithm can exhibit markedly different behaviors, such as linear vs. quadratic convergence, depending on initialization.*

We consider matrix factorization as a canonical example, where the goal is to solve i) symmetric problems, $\min_{\mathbf{X}} \|\mathbf{X}\mathbf{X}^\top - \mathbf{A}\|_F^2$; and ii) asymmetric ones, $\min_{\mathbf{X},\mathbf{Y}} \|\mathbf{X}\mathbf{Y}^\top - \mathbf{A}\|_F^2$. While these classical problems can be handled via various approaches, they are notoriously challenging for optimization, since they are nonconvex, nonsmooth (albeit differentiable), non-coercive (for asymmetric problems), and do not satisfy Polyak-Lojasiewicz (PL) condition (Chi et al., 2019). Let $\mathbf{A} \in \mathbb{R}^{m \times n}$ (or $\mathbf{A} \in \mathbb{R}^{m \times m}$) for asymmetric (symmetric) problems, $\mathbf{X} \in \mathbb{R}^{m \times r}$ and $\mathbf{Y} \in \mathbb{R}^{n \times r}$. Building on the relation of rank($\mathbf{A}$) and $r$, we can categorize matrix factorization into three setups: exact-parametrized (rank($\mathbf{A}$) = $r$), over-parametrized (rank($\mathbf{A}$) < $r$), and under-parametrized (rank($\mathbf{A}$) > $r$).

The asymmetric problem ii) is thoroughly explored in the literature. For the exact- and over-parametrized cases, global convergence has been established for GD, Alternating GD (AltGD), and ScaledGD (Du et al., 2018; Ye & Du, 2021; Ward & Kolda, 2023; Jia et al., 2023; Tong et al., 2021), where most of them admit a linear rate. Regarding under-parametrized settings, only asymptotic global convergence of GD is established in (Du et al., 2018) to the best of our knowledge. Common to above algorithms is the small initialization with $\mathbf{X}_0 \sim \mathcal{N}(0, \zeta_x^2)$ and $\mathbf{Y}_0 \sim \mathcal{N}(0, \zeta_y^2)$ for some sufficiently small $\zeta_x^2$ and $\zeta_y^2$. However, such initialization results in unfavorable performance both theoretically and empirically, partly because of the need of escaping from a saddle point $(\mathbf{0}, \mathbf{0})$.

This work proposes *Nyström initialization* to effectively bypass the aforementioned saddle point. More importantly, it significantly enhances the global convergence rates when applied on top of ScaledGD. In the exact- and over-parametrized settings, Nyström initialization boosts ScaledGD to converge at a *quadratic* rate (i.e., $\mathcal{O}(\log\log(1/\epsilon))$) on symmetric problems and enables a *one-step* convergence for asymmetric problems. For the more challenging case with under-parametrization, we prove that with our Nyström initialization, ScaledGD converges at a linear rate to the neighbor of

Table 1: Comparison of complexity for global optimality for (a)symmetric matrix factorization. Here, EP, OP, and UP are abbreviations for exact-, over- and under- parametrization. The prescribed optimality error is denoted as $\epsilon$, and $\kappa$ is the condition number of $\mathbf{A}$. Note that our bounds for UP depict the complexity to near optima. The "special" initialization in AltGD is still a small initialization, but with more careful designs that will be clear in Sec. 3.1. Works marked with * are for the closely related matrix sensing setting (hence the comparison may not be fair).

| setting | | alg. | ref. | init. | rate |
|---|---|---|---|---|---|
| Asymmetric | EP | GD | (Ye & Du, 2021) | small | $\mathcal{O}\big(\kappa^4 + \kappa^4 \log(1/\epsilon)\big)$ |
| | | AltGD | (Ward & Kolda, 2023) | special | $\mathcal{O}\big(\kappa^2 \log(1/\epsilon)\big)$ |
| | | ScaledGD | (Tong et al., 2021) | local | $\mathcal{O}(\log(1/\epsilon))$ |
| | | ScaledGD | **Theorem 3** | Nyström | $\mathcal{O}(1)$ |
| | OP | modified GD* | (Xiong et al., 2024) | small | $\mathcal{O}(\kappa^2 \log(1/\epsilon))$ |
| | | AltGD | (Ward & Kolda, 2023) | special | $\mathcal{O}\big(\kappa^2 \log(1/\epsilon)\big)$ |
| | | ScaledGD | **Theorem 6** | Nyström | $\mathcal{O}(1)$ |
| | UP | GD | (Du et al., 2018) | small | asymptotic |
| | | ScaledGD | **Theorem 4** | Nyström | $\mathcal{O}(1)$ |
| Symmetric | EP | GD* | (Stöger & Soltanolkotabi, 2021) | small | $\mathcal{O}\big(\kappa^8 + \kappa^2 \log(1/\epsilon)\big)$ |
| | | ScaledGD | **Theorem 1** | Nyström | $\mathcal{O}\big(\kappa^3 + \log\log(1/\epsilon)\big)$ |
| | OP | GD* | (Stöger & Soltanolkotabi, 2021) | small | $\mathcal{O}\big(\kappa^8 + \kappa^6 \log(\kappa/\epsilon)\big)$ |
| | | ScaledGD-$\lambda$* | (Xu et al., 2023) | small | $\mathcal{O}\big(\log^2 \kappa + \log(1/\epsilon)\big)$ |
| | | ScaledGD | **Theorem 5** | Nyström | $\mathcal{O}\big(\kappa^3 + \log\log(1/\epsilon)\big)$ |
| | UP | ScaledGD | **Theorem 2** | Nyström | $\mathcal{O}(1/\epsilon \cdot \log(1/\epsilon))$ |

a global optimum on symmetric problems, and then exhibits a sublinear rate to a more fine-grained neighboring area. Overall, Nyström initialization enables us to improve existing rates in exact-, over-, and under-parametrized settings; see more detailed comparisons in Tab. 1.

Our results highlight that the convergence of ScaledGD is *critically determined by the initialization*. Taking symmetric and exact-parametrized problems as an example, our quadratic rate slows down to a linear one when adopting either small initialization or slightly perturbed Nyström initialization.

After demonstrating the theoretical merits of Nyström initialization, we further extend its applications to another scenario with Burer-Monteiro factorization, in the context of LoRA for finetuning deep neural networks (Hu et al., 2022). This is motivated by the fact that asymmetric matrix factorization is equivalent to LoRA applied on linear models with whitened data (Arora et al., 2018), and is in line with several recent works that take insights from matrix factorization to improve LoRA (Zhang & Pilanci, 2024; Yaras et al., 2024). Compared with existing strategies for initializing LoRA (Büyükakyüz, 2024; Meng et al., 2024; Wang et al., 2024), our Nyström initialization for LoRA (abbreviated as NoRA) is more economical and aligns better with existing deployment pipelines. The effectiveness of NoRA is demonstrated on downstream tasks from various domains, through both diffusion and large language models (LLMs). In a nutshell, our contributions can be summarized as:

❖ **Faster rates.** Nyström initialization is provably beneficial to ScaledGD. For symmetric problems, it catalyzes not only the first *quadratic rate* in exact- and over- parameterized settings, but also a (sub)linear rate for under-parametrization where only asymptotic results were known. It also allows more remarkable improvement on asymmetric problems; see details in Tab. 1.

❖ **Critical role of initialization.** Our theoretical results convey an intriguing message for nonconvex (nonsmooth) optimization: the behaviors of the same algorithm, whether converging at a quadratic or linear rate, are critically determined by initialization.

❖ **Practical implications.** We further illustrate the power of Nyström initialization for finetuning diffusion and large language models (LLMs). The resultant approach, NoRA, effectively improves the performance of LoRA on several representative tasks.

**Notation**. Bold lowercase (capital) letters denote column vectors (matrices); $(\cdot)^\top$, $(\cdot)^\dagger$ and $\|\cdot\|_\mathsf{F}$ refer to transpose, pseudo inverse, and Frobenius norm of a matrix; $\|\cdot\|$ is the $\ell_2$ (spectrum) norm of a vector (matrix); $\sigma_i(\cdot)$ and $\lambda_i(\cdot)$ denote the $i$-th largest singular value and eigenvalue, respectively.

## 1.1 RELATED WORKS

We only streamline results on the convergence of matrix factorization under the broad umbrella of quartic optimization. Other closely related topics, such as LoRA variants, can be found in Apdx. A.1.

**Quartic Optimization.** Matrix factorization problems considered in this work are classical examples of forth-order growth functions. It involves a complex landscape characterized by nonconvexity, nonsmoothness, and the absence of PL condition. Similar to other works listed in Tab. 1, the goal of this work is to recap this classical problem and to unveil intriguing behaviors from an optimization perspective. Recent works have examined the convergence of several algorithms, such as GD, AltGD, and ScaledGD (Du et al., 2018; Ye & Du, 2021; Ward & Kolda, 2023; Jia et al., 2023; Jiang et al., 2023b) in exact- and over- parametrized settings. Most of them admit linear convergence with different dependences on the condition number of the factorized matrix $\mathbf{A}$. A concurrent work (Xu et al., 2024) studies Nyström initialization for Nesterov's accelerated method on the asymmetric and over-parametrized setting, giving an improved condition number dependence over GD. GD for matrix square root problems is studied in (Jain et al., 2017). Another closely related setting within forth-order growth is matrix sensing; see e.g., (Stöger & Soltanolkotabi, 2021; Zhang et al., 2021; Jin et al., 2023; Xiong et al., 2024). Linear rates are obtained for problems with exact- and over-parametrization, despite some of them demand early stopping. Similar to matrix factorization, not too much is known for under-parametrization. There are other approaches to tackle general forth-order growth optimization. For example, relative smoothness is considered in (Lu et al., 2018); adaptive step sizes induced by fine-grained geometry are studied in (Davis et al., 2024). The work of (Dragomir & Nesterov, 2023) also copes with such problems but requires convexity of the objective.

## 2 THE POWER OF INITIALIZATION FOR SYMMETRIC MATRIX FACTORIZATION

## 2.1 PRELIMINARIES

We start to examine the critical role of initialization on symmetric matrix factorization problems

$$\min_{\mathbf{X}\in\mathbb{R}^{m\times r}} \frac{1}{4}\|\mathbf{X}\mathbf{X}^\top - \mathbf{A}\|_\mathsf{F}^2. \tag{1}$$

Within this section, we assume that $\mathbf{A} \in \mathbb{R}^{m\times m}$ is positive semidefinite (PSD), otherwise one can employ the asymmetric formulation as in later sections. Problem (1) also closely links with matrix sensing, particularly under a sufficient number of Gaussian measurements (Xiong et al., 2024). From an optimization perspective, problem (1) is nonconvex and has no global Lipschitz gradient (Tu et al., 2016; Chi et al., 2019). These undesirable properties pose challenges for analyzing its convergence.

Notationally, let $r_A := \mathrm{rank}(\mathbf{A})$ and further denote the compact eigendecomposition as $\mathbf{A} = \mathbf{Q}\mathbf{\Sigma}\mathbf{Q}^\top$, where $\mathbf{Q} \in \mathbb{R}^{m\times r_A}$ and $\mathbf{\Sigma} \in \mathbb{R}^{r_A \times r_A}$. Since PSD matrices share the same eigen and singular values, we employ $\sigma_i(\cdot)$ to denote both in this section. Without loss of generality, we assume that the largest and smallest singular values are $\sigma_1(\mathbf{A}) = 1$ and $\sigma_{r_A}(\mathbf{A}) = 1/\kappa$ such that the condition number is $\kappa$.

**ScaledGD as our optimizer.** We investigate the power of initialization on ScaledGD (Tong et al., 2021), a preconditioned version of GD; see detailed discussions in e.g., (Tong et al., 2021; Jia et al., 2023). Starting from $t = 0$, the update of ScaledGD is given by

$$\mathbf{X}_{t+1} = \mathbf{X}_t - \eta(\mathbf{X}_t\mathbf{X}_t^\top - \mathbf{A})\mathbf{X}_t \cdot (\mathbf{X}_t^\top\mathbf{X}_t)^{-1}. \tag{2}$$

The inversion of the $r \times r$ matrix $\mathbf{X}_t^\top\mathbf{X}_t$ is computationally feasible in the low-rank setting with $r \ll m$. Small initialization is widely adopted, i.e., $[\mathbf{X}_0]_{ij} \sim \mathcal{N}(0, \zeta^2)$, where $\zeta$ is a sufficiently small positive number. Under such initialization, ScaledGD converges linearly for exact-parametrization $(r = r_A)$,[1] yet less is known for under- and over-parametrization; see more in Tab. 1. Next, we show that a simple yet effective initialization can provoke faster convergence of ScaledGD.

---

[1]This linear rate is indicated by our numerical results in Fig. 1 (a). While we are not aware of a direct proof for this observation, it is presumable that the analysis in the asymmetric and exact-parametrized setting (Jia et al., 2023) could be adapted to provide some guarantees.

## 2.2 Nyström initialization

To improve the convergence rates, it is essential to ensure that the initialization satisfies two conditions for exact- and under-parametrized problems[2]: i) each column of $\mathbf{X}_0$ is in the column space of $\mathbf{A}$, and ii) $\mathbf{X}_0$ is full rank, i.e., rank$(\mathbf{X}_0) = r$. The analytical rationale will be elucidated in the subsequent sections. A straightforward means to meet these conditions is via Nyström sketch (Tropp et al., 2017)

$$\textbf{Nyström initialization:} \quad \mathbf{X}_0 = \mathbf{A}\boldsymbol{\Omega}, \quad \text{where } [\boldsymbol{\Omega}]_{ij} \sim \mathcal{N}(0, \xi^2), \forall i, \forall j \qquad (3)$$

where $\boldsymbol{\Omega} \in \mathbb{R}^{m \times r}$ is a Gaussian random matrix. Note that the Gaussian $\boldsymbol{\Omega}$ is only for practical convenience, and it does not exclude other valid choices such as a random draw from a Stiefel manifold St$(m, r)$. From the initialization in (3), it is not difficult to see that condition i) is satisfied already. Our next lemma shows that condition ii) holds w.h.p.

**Lemma 1** (Initialization for exact- and under- parametrization). *For some universal constant $\tau > 0$, $\sigma_r(\mathbf{X}_0) \geq \xi\tau(\sqrt{r_A} - \sqrt{r-1})\sigma_{r_A}(\mathbf{A})$ is satisfied with high probability, i.e., rank$(\mathbf{X}_0) = r$ w.h.p.*

The detailed expression for this "high probability" in Lemma 1 can be found in Apdx. B.1.1. Note that although we do not state explicitly, most of our results below hold under rank$(\mathbf{X}_0) = r$ in exact- and under-parametrized setting, while rank$(\mathbf{X}_0) = r_A$ is needed when over-parametrized.

## 2.3 Nyström initialization in the exact-parametrized setting

We start with Nyström initialization for exact-parametrized problems, i.e., $r_A = r$. Our first result dives into the implicit regularization induced by the ScaledGD under the proposed initialization.

**Lemma 2.** *If $\mathbf{X}_0$ is obtained by Nyström initialization (3), ScaledGD in (2) ensures that for all $t \geq 0$*

*i) every column of $\mathbf{X}_t$ is in the column space of $\mathbf{A}$, and $\mathbf{X}_t = \mathbf{Q}\boldsymbol{\Phi}_t$ for some $\boldsymbol{\Phi}_t \in \mathbb{R}^{r \times r}$; and,*

*ii) the smallest eigenvalue of $\mathbf{X}_t\mathbf{X}_t^\top$ satisfies that*

$$\sigma_r(\mathbf{X}_{t+1}\mathbf{X}_{t+1}^\top) \geq (1-\eta)^{2t+2}\sigma_r(\mathbf{X}_0\mathbf{X}_0^\top) + (1-\eta)\sigma_r(\mathbf{A}) - (1-\eta)^{2t+3}\sigma_r(\mathbf{A}).$$

Lemma 2 implies the full rankness of $\mathbf{X}_t$ over the trajectory, i.e., rank$(\mathbf{X}_t) = $ rank$(\boldsymbol{\Phi}_t) = r, \forall t$. This ensures an invertible preconditioner $\mathbf{X}_t^\top \mathbf{X}_t$. In other words, iteration (2) is well-defined. The most important implication of Lemma 2 is the alignment of $\mathbf{X}_t$ with the directions of eigenvectors of $\mathbf{A}$, that is, $\mathbf{X}_t = \mathbf{Q}\boldsymbol{\Phi}_t$. This can be equivalently understood as the elimination of the residual space, i.e., $(\mathbf{I} - \mathbf{Q}\mathbf{Q}^\top)\mathbf{X}_t = \mathbf{0}, \forall t$. While we will expand this discussion shortly, this alignment in directions enables us to establish a quadratic rate for ScaledGD.

**Theorem 1.** *With Nyström initialization (3), ScaledGD in (2) exhibits a two-phase behavior.*

*Phase 1 (linear convergence). Let $\eta = \mathcal{O}(\frac{1}{\kappa^3\|\mathbf{A}\|_F})$. After $T_1 := \mathcal{O}(\kappa^3\sqrt{r}\log\kappa)$ iterations, ScaledGD ensures that $\|\mathbf{X}_{T_1}\mathbf{X}_{T_1}^\top - \mathbf{A}\|_F \leq \mathcal{O}(1/\kappa^2)$; and,*

*Phase 2 (quadratic convergence). After Phase I, ScaledGD converges quadratically with $\eta = 0.5$. In particular, $\|\mathbf{X}_T\mathbf{X}_T^\top - \mathbf{A}\|_F \leq \epsilon$ is achieved after $T = \mathcal{O}\left(\log\log(\frac{1}{\kappa\epsilon})\right)$ iterations.*

Theorem 1 establishes that global optimality of (1) is attained by ScaledGD within $\mathcal{O}(\kappa^3\sqrt{r}\log\kappa + \log\log\frac{1}{\kappa\epsilon})$ iterations. ScaledGD first converges to a local region satisfying $\|\mathbf{X}_t\mathbf{X}_t^\top - \mathbf{A}\|_F \leq \mathcal{O}(\frac{1}{\kappa^2})$ linearly, after which a quadratic rate can be granted. This is, to the best of our knowledge, the first quadratic rate for symmetric matrix factorization (1). Interestingly, it is achieved without requiring (exact) Hessian on a nonconvex and nonsmooth problem. A graphical illustration of this quadratic rate can be found in Fig. 1 (a) using synthetic data detailed in Apdx. E.1. It is observed that ScaledGD with Nyström initialization outperforms linearly converging algorithms such as GD and ScaledGD with small initialization. Moreover, it is worth emphasizing that Theorem 1 has no requirement on the magnitude of Nyström initialization – it does not need $\xi$ in (3) to be small. Compared with a small initialization, i.e., $\mathbf{X}_0 \approx \mathbf{0}$, this avoids escaping from the stationary point $\mathbf{0}$. The convergence of ScaledGD under various choices of $\xi$ can be found in (the solid lines of) Fig. 1(b).

---

[2]For the ease of presentation, the over-parametrized setting is considered in the appendix.

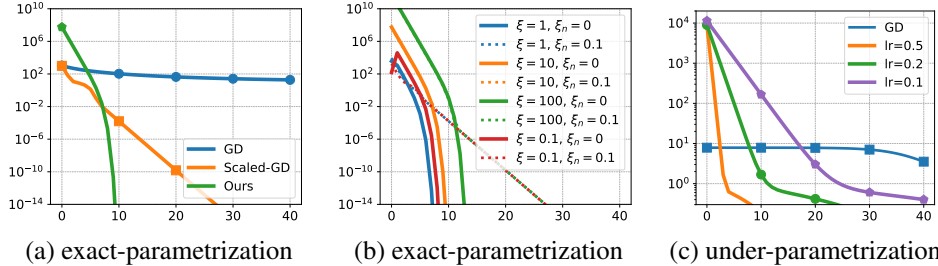

(a) exact-parametrization     (b) exact-parametrization     (c) under-parametrization

Figure 1: Convergence of ScaledGD under Nyström initialization (optimality error vs. iteration) in different settings. (a) Comparison of GD, and ScaledGD with small / Nyström initialization (ours). (b) Solid lines show that our initialization is not sensitive to magnitude of $\xi$; and dotted lines illustrate that quadratic convergence cannot be obtained after perturbing the initialization, i.e., $\mathbf{X}_0 = \mathbf{A}\boldsymbol{\Omega} + \mathbf{N}$, where $[\mathbf{N}]_{ij} \sim \mathcal{N}(0, \xi_n^2)$. (c) Comparison of ScaledGD under Nyström initialization with various $\eta$.

**The critical role of initialization.** As shown in Lemma 2, Nyström initialization aligns $\mathbf{X}_t$ to the directions of eigenvectors $\mathbf{Q}$, thereby eliminating the residual space, i.e., $(\mathbf{I} - \mathbf{Q}\mathbf{Q}^\top)\mathbf{X}_t = \mathbf{0}, \forall t$. This is critical for a quadratic rate and it is in stark contrast with most of existing works (Du et al., 2018; Ye & Du, 2021; Jia et al., 2023), where small initialization only guarantees that $\|(\mathbf{I} - \mathbf{Q}\mathbf{Q}^\top)\mathbf{X}_t\|_{\mathsf{F}}$ converges to 0 at a linear rate. We graphically illustrate this point in Fig. 1 (b), where we perturb Nyström initialization slightly to inject noise into the residual space. Reflected in the dotted lines, even if the noise is so small such that the earlier convergence does not differ from Nyström initialization, only a linear convergence can be observed for the perturbed initialization.

**Extensions to the case of over-parametrization.** Nyström initialization is further extended to cope with over-parametrized case ($r > r_A$) in Apdx. B.4. For this specific setup, we slightly modify ScaledGD by substituting the possibly non-invertible $(\mathbf{X}_t^\top \mathbf{X}_t)^{-1}$ in (2) with $(\mathbf{X}_t^\top \mathbf{X}_t)^\dagger$; see (26). Unlike previous works (Xu et al., 2023; Zhang et al., 2021), our modification requires no damping parameters thanks to our Nyström initialization. This leads to, as far as we know, the first quadratic rate for over-parametrized problems. Additional numerical experiments on over-parametrized problems are provided in Fig. 4 in appendix to validate the established quadratic rate.

## 2.4 NYSTRÖM INITIALIZATION IN THE UNDER-PARAMETRIZED SETTING

Next, we consider the under-parametrized case of (1), i.e., $r < r_A$. To the best of our knowledge, only asymptotic convergence is established for GD on such problems (Du et al., 2018). This is partially because that even the local PL condition is challenging to be verified. With Nyström initialization, we will show that ScaledGD converges under a slightly weaker criterion.

**Definition 1** (Weak optimality). *Matrix $\mathbf{X} \in \mathbb{R}^{m \times r}$ is weakly optimal to* (1) *if* $\mathbf{X}^\top \mathbf{A}^\dagger \mathbf{X} - \mathbf{I}_r = \mathbf{0}$.

Our first result characterizes that all global optima are also weakly optimal. In other words, if weak optimality is ensured, this algorithm has a chance to reach a global optimum as well.

**Lemma 3.** *All globally optimal solutions to* (1) *are also weakly optimal.*

We then focus on the convergence of ScaledGD to weak optimality. In the case of under-parametrization, Nyström initialization also aligns $\mathbf{X}_t$ to the directions of eigenvectors of $\mathbf{A}$.

**Lemma 4.** *If ScaledGD in* (2) *is equipped with Nyström initialization* (3)*, one can write* $\mathbf{X}_t = \mathbf{Q}\boldsymbol{\Phi}_t, \forall t$ *for some* $\boldsymbol{\Phi}_t \in \mathbb{R}^{r_A \times r}$.

Lemma 4 shows that $(\mathbf{I} - \mathbf{Q}\mathbf{Q}^\top)\mathbf{X}_t = \mathbf{0}, \forall t$ also holds, namely, Nyström initialization eliminates the residual space. Building upon this, the convergence of ScaledGD can be established.

**Theorem 2.** *The following holds for ScaledGD* (2) *with Nyström initialization* (3):

*i) (Linear convergence to neighborhood of weak optima). If one chooses a constant $\eta \leq 1$, ScaledGD ensures that $\|\mathbf{X}_t^\top \mathbf{A}^\dagger \mathbf{X}_t - \mathbf{I}_r\|_{\mathsf{F}} \leq \mathcal{O}(\eta r) + \epsilon$ in $\mathcal{O}(\log \frac{1}{\epsilon})$ iterations; or,*

*ii) (Convergence to weak optima). Let $\eta = \mathcal{O}(\epsilon/r)$, weak optimality is ensured by ScaledGD after $\mathcal{O}(\frac{r}{\epsilon} \log \frac{1}{\epsilon})$ iterations, i.e., $\|\mathbf{X}_t^\top \mathbf{A}^\dagger \mathbf{X}_t - \mathbf{I}_r\|_{\mathsf{F}} \leq \epsilon$.*

If one chooses a constant learning rate e.g, $\eta = 0.1$, linear convergence can be established until reaching a neighboring area of a weakly optimal solution. The error $\|\mathbf{X}_t^\top \mathbf{A}^\dagger \mathbf{X}_t - \mathbf{I}_r\|_\mathsf{F} = \mathcal{O}(\eta r)$ is low, given that $r$ is typically small in practice. A graphical illustration of this linear rate can be found in Fig. 1 (c). On the other hand, if the learning rate is chosen according to the prescribed accuracy $\epsilon$, one can obtain a sublinear rate $\mathcal{O}(\frac{r}{\epsilon} \log \frac{1}{\epsilon})$ to exact weak optimality. These behaviors clearly indicate a step scheduling of learning rates (e.g., setting $\eta = 0.1, 0.01, \ldots$ every a few iterations) for both fast convergence and exact weak optimality in practice. It is also worth mentioning that the convergence under both choices of $\eta$ has no dependence on $\kappa$. This aligns with the presumption in previous works (Tong et al., 2021; Jia et al., 2023) that ScaledGD performs well on ill-conditioned problems, providing the first rigorous justification for the under-parametrized setting.

Finally, we show that even in the worst case, ScaledGD guarantees that $\mathbf{X}_t$ converges to a point that is adequately close to a global solution, and the relative distance is sublinear in $r$.

**Lemma 5.** *Let $\mathbf{Q}_1$ be the first $r$ column on $\mathbf{Q}$, and $\mathbf{\Sigma}_1$ be the top-left $r \times r$ sub-block of $\mathbf{\Sigma}$. Denote an optimal solution to* (1) *as $\mathbf{X}_* = \mathbf{Q}_1 \mathbf{\Sigma}_1^{1/2}$. ScaledGD* (2) *with Nyström initialization* (3) *ensures*

$$\lim_{t \to \infty} \|\mathbf{X}_t - \mathbf{X}_*\|_\mathsf{F} \leq \mathcal{O}(r^{3/4}).$$

## 3 THE POWER OF INITIALIZATION FOR ASYMMETRIC MATRIX FACTORIZATION

### 3.1 INITIALIZATION AND MODIFIED SCALEDGD

This section demonstrates that the power of initialization is even more striking in solving asymmetric matrix factorization than symmetric ones. Given $\mathbf{A} \in \mathbb{R}^{m \times n}$, consider the following problem

$$\min_{\mathbf{X} \in \mathbb{R}^{m \times r}, \mathbf{Y} \in \mathbb{R}^{n \times r}} \frac{1}{2} \|\mathbf{X}\mathbf{Y}^\top - \mathbf{A}\|_\mathsf{F}^2. \tag{4}$$

Denote $\text{rank}(\mathbf{A}) = r_A$, and the compact SVD as $\mathbf{A} = \mathbf{U}\mathbf{\Sigma}\mathbf{V}^\top$, where $\mathbf{U} \in \mathbb{R}^{m \times r_A}, \mathbf{\Sigma} \in \mathbb{R}^{r_A \times r_A}$, and $\mathbf{V} \in \mathbb{R}^{n \times r_A}$. Similar to the previous section, we assume that $\sigma_1(\mathbf{A}) = 1$ and $\sigma_{r_A}(\mathbf{A}) = 1/\kappa$.

**Nyström initialization.** We adopt an asymmetric manner to initialize $\mathbf{X}_0$ and $\mathbf{Y}_0$ for (4), i.e.,

$$\boxed{\text{Nyström initialization:} \quad \mathbf{X}_0 = \mathbf{A}\mathbf{\Omega}, \quad \mathbf{Y}_0 = \mathbf{0}} \tag{5}$$

where $\mathbf{\Omega}$ is a Gaussian random matrix of $\mathbb{R}^{n \times r}$ with $[\mathbf{\Omega}]_{ij} \sim \mathcal{N}(0, \xi^2), \forall i, \forall j$. We can follow the same steps of Lemma 1 to show that $\mathbf{X}_0$ in (5) is rank $r$ w.h.p. in exact- and under-parametrized settings. Moreover, there is no requirement on the magnitude of $\xi$, meaning that it is possible to start far from the saddle point $(\mathbf{0}, \mathbf{0})$. This asymmetry of $\mathbf{X}_0$ and $\mathbf{Y}_0$ in (5) is in contrast with small initialization which typically induces $\|\mathbf{X}_0\|_\mathsf{F} \approx \|\mathbf{Y}_0\|_\mathsf{F}$ (Du et al., 2018; Jia et al., 2023). The merits will become clear shortly. Note that AltGD (Ward & Kolda, 2023) also adopts sketch at initialization, i.e., $\mathbf{X}_0 = \mathcal{O}(\mathbf{A}\mathbf{\Omega}_1/\sigma_1(\mathbf{A}))$ and $\mathbf{Y}_0 = \mathcal{O}(\sigma_1(\mathbf{A})\mathbf{\Omega}_2)$, where $\mathbf{\Omega}_1$ and $\mathbf{\Omega}_2$ are Gaussian random matrices. Besides the requirement on small variance of $\mathbf{\Omega}_1$ and $\mathbf{\Omega}_2$ and the explicit need of $\sigma_1(\mathbf{A})$, this initialization cannot eliminate the residual space. Consequently, AltGD demands early stopping in exact- and over-parametrized problems, and little is known for under-parametrized case.

**Modified ScaledGD.** To adapt to the non-invertible $\mathbf{Y}_0^\top \mathbf{Y}_0 = \mathbf{0}$ in Nyström initialization (5), we modify the first iteration of ScaledGD. More precisely, the updates are summarized below

$$\mathbf{X}_1 = \mathbf{X}_0, \text{ and } \mathbf{X}_{t+1} = \mathbf{X}_t - \eta(\mathbf{X}_t\mathbf{Y}_t^\top - \mathbf{A})\mathbf{Y}_t(\mathbf{Y}_t^\top\mathbf{Y}_t)^{-1}, \forall t \geq 1; \tag{6a}$$

$$\mathbf{Y}_{t+1} = \mathbf{Y}_t - \eta(\mathbf{X}_t\mathbf{Y}_t^\top - \mathbf{A})^\top \mathbf{X}_t(\mathbf{X}_t^\top\mathbf{X}_t)^{-1}, \forall t \geq 0. \tag{6b}$$

### 3.2 NYSTRÖM INITIALIZATION IN THE EXACT-PARAMETRIZED SETTING

We start with the exact-parametrized case, i.e., $r_A = r$ in (4). The benefit of Nyström initialization (5) for iteration (6) is again the alignment of $\mathbf{X}_t$ and $\mathbf{Y}_t$ to the directions of singular vectors.

**Lemma 6.** *The modified ScaledGD in* (6) *under Nyström initialization* (5) *guarantees that $\mathbf{X}_t = \mathbf{U}\mathbf{\Phi}_t$ and $\mathbf{Y}_t = \mathbf{V}\mathbf{\Psi}_t$, $\forall t \geq 0$ for some $\mathbf{\Phi}_t \in \mathbb{R}^{r \times r}$ and $\mathbf{\Psi}_t \in \mathbb{R}^{r \times r}$.*

Similar to the symmetric problems, the implication of Lemma 6 is the elimination of residual space, i.e., $(\mathbf{I} - \mathbf{U}\mathbf{U}^\top)\mathbf{X}_t = \mathbf{0}$ and $(\mathbf{I} - \mathbf{V}\mathbf{V}^\top)\mathbf{Y}_t = \mathbf{0}$. This turns out to be even more beneficial for asymmetric problems, as it induces one-step convergence of ScaledGD.

**Theorem 3** (One-step convergence). *With $\eta = 1$ and Nyström initialization* (5)*, the modified ScaledGD* (6) *guarantees $\mathbf{X}_1\mathbf{Y}_1^\top = \mathbf{A}$. In other words, global convergence is achieved in one step.*

Comparing to symmetric case (cf. Theorem 1), Theorem 3 suggests that solving problem (4) requires less iterations owing to the asymmetry of $\mathbf{X}_0$ and $\mathbf{Y}_0$ at initialization (5). This agrees with results in (Xiong et al., 2024), which illustrate the benefit of asymmetry for matrix sensing.

**Remark 1.** *The one step convergence of ScaledGD can also be interpreted as a new approach for matrix factorization. While the complexity of this factorization is generally equivalent to that of SVD, it can be faster than SVD when $\mathcal{O}(n^{0.38}) < r \leq n$. Further details are provided in Apdx. A.5.*

Lastly, we present a result that may be of independent interest – the asymmetric and symmetric problems are interconnected under our Nyström initialization. This link is made clear in the proof of the following corollary (to Theorem 1), which states that ScaledGD admits quadratic convergence under different choices of step sizes.

**Corollary 1** (Quadratic convergence). *With Nyström initialization* (5) *and different choices of step sizes, modified ScaledGD in* (6) *has a similar behavior as Theorem 1, i.e.,*

*Phase 1 (linear convergence). Let $\eta = \mathcal{O}(\frac{1}{\kappa^3\|\mathbf{A}\|_\mathsf{F}})$. After $T_1 := \mathcal{O}(\kappa^3\sqrt{r}\log\kappa)$ iterations, ScaledGD ensures that $\|\mathbf{X}_{T_1}\mathbf{Y}_{T_1}^\top - \mathbf{A}\|_\mathsf{F} \leq \mathcal{O}(1/\kappa^2)$.*

*Phase 2 (quadratic convergence). After Phase I, ScaledGD converges quadratically with $\eta = 0.5$. In particular, $\|\mathbf{X}_T\mathbf{Y}_T^\top - \mathbf{A}\|_\mathsf{F} \leq \epsilon$ is ensured after $T = \mathcal{O}\big(\log\log(\frac{1}{\kappa\epsilon})\big)$ iterations.*

**Extensions to over-parametrization.** One-step global convergence can also be established for over-parametrized asymmetric problems under Nyström initialization. More on this can be found in Apdx. C.3, where we provide the first convergence result on ScaledGD under such a setup.

### 3.3 NYSTRÖM INITIALIZATION IN THE UNDER-PARAMETRIZED SETTING

Lastly, we tackle the case of under-parametrization in the asymmetric problem (4), where $r_A > r$. Similar to the symmetric case in Sec.2.4, we consider a slightly weaker version of optimality.

**Definition 2** (Generalized weak optimality). *We say $(\mathbf{X}, \mathbf{Y})$ is weakly optimal if $\mathbf{Y}^\top\mathbf{A}^\dagger\mathbf{X} - \mathbf{I}_r = \mathbf{0}$.*

Generalized weak optimality is satisfied by any global optimum, which is proved in Lemma 13 in the appendix. With this preparation, we are ready to show that ScaledGD converges in a single step.

**Theorem 4.** *If $\eta = 1$, ScaledGD in* (6) *with Nyström initialization* (5) *ensures generalized weak optimality in one iteration, i.e., $\mathbf{Y}_1^\top\mathbf{A}^\dagger\mathbf{X}_1 - \mathbf{I}_r = \mathbf{0}$.*

**The critical role of initialization.** Through the theoretical analyses in the previous two sections, it is evident that the convergence of ScaledGD for matrix factorization is *highly dependent on the initialization*. Here is an intuitive, though not strictly rigorous, summary: Small initialization results in behaviors similar to first-order optimizers, i.e., linear convergence (Jia et al., 2023). In contrast, the proposed Nyström initialization catalyzes quadratic rates and even one-step convergence, resembling the optimization trajectory of second-order approaches such as Newton's method (Nesterov, 2004).

## 4 NORA: NYSTRÖM LOW RANK ADAPTERS

Our theoretical results highlight the merits of suitable initialization for matrix factorization problems. One of the key insights is that the Burer-Monterio factorization benefits from good directions of $\mathbf{X}_0$ and $\mathbf{Y}_0$ at initialization; cf. Lemmas 2 and 6. We term this as *directional alignment*. In this section, we extend the benefit of initialization to practical scenarios, showing that directional alignment is also beneficial for low-rank adapters (LoRA) in finetuning deep neural networks (Hu et al., 2022).

LoRA enhances parameter efficiency of finetuning by approximating the unknown parameter-change $\Delta\mathbf{W} \in \mathbb{R}^{m \times n}$ through Burer-Monterio factorization

$$\mathbf{W}_0 + \Delta\mathbf{W} \approx \mathbf{W}_0 + \mathbf{X}\mathbf{Y}^\top \tag{7}$$

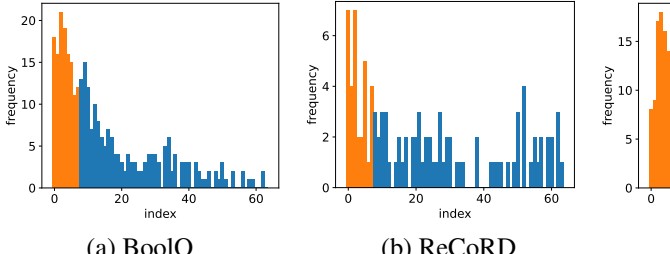

|  (a) BoolQ | (b) ReCoRD | (c) RTE |

Figure 2: Which singular values have the largest change after finetuning with LoRA of rank $r$? Orange: top-$r$ singular values; blue: other singular values. Note that here we only plot the first 64 singular values as others rarely have sufficiently large change.

Table 2: Performance of NoRA and NoRA+ for few-shot learning with OPT-1.3B.

| OPT-1.3B | SST-2 | WSC | BoolQ | CB | RTE | ReCoRD | MultiRC | SQuAD | avg ($\uparrow$) |
|---|---|---|---|---|---|---|---|---|---|
| Prefix | $92.9_{\pm0.9}$ | $59.6_{\pm1.6}$ | $73.1_{\pm2.3}$ | $71.6_{\pm2.9}$ | $65.2_{\pm2.6}$ | $69.7_{\pm1.0}$ | $64.4_{\pm3.2}$ | $82.2_{\pm1.4}$ | 72.3 |
| LoRA | $93.1_{\pm0.2}$ | $59.1_{\pm2.0}$ | $70.6_{\pm5.2}$ | $72.6_{\pm3.7}$ | $69.1_{\pm4.7}$ | $70.8_{\pm1.0}$ | $68.0_{\pm1.4}$ | $81.9_{\pm1.8}$ | 73.2 |
| OLoRA | $92.7_{\pm0.5}$ | $60.0_{\pm2.3}$ | $70.9_{\pm3.1}$ | $80.3_{\pm2.7}$ | $69.7_{\pm1.0}$ | $71.3_{\pm1.2}$ | $66.7_{\pm0.9}$ | $80.0_{\pm1.4}$ | 74.0 |
| PiSSA | $92.7_{\pm0.6}$ | $60.6_{\pm3.7}$ | $70.4_{\pm0.7}$ | $78.0_{\pm7.2}$ | $70.4_{\pm2.8}$ | $70.9_{\pm1.2}$ | $67.9_{\pm2.1}$ | $82.1_{\pm0.4}$ | 74.1 |
| **NoRA** | $93.4_{\pm0.7}$ | $60.6_{\pm3.8}$ | $73.2_{\pm0.6}$ | $79.2_{\pm5.2}$ | $72.0_{\pm1.3}$ | $71.3_{\pm1.0}$ | $68.5_{\pm1.2}$ | $81.8_{\pm0.7}$ | **75.0** |
| **NoRA+** | $93.2_{\pm0.5}$ | $61.2_{\pm0.6}$ | $72.9_{\pm1.3}$ | $79.5_{\pm5.8}$ | $72.4_{\pm3.6}$ | $71.5_{\pm0.9}$ | $68.4_{\pm1.2}$ | $82.0_{\pm0.9}$ | **75.1** |

where $\mathbf{W}_0 \in \mathbb{R}^{m \times n}$ is the pretrained weight (of a particular layer), and $\mathbf{X} \in \mathbb{R}^{m \times r}$ and $\mathbf{Y} \in \mathbb{R}^{n \times r}$ with $r \ll \min\{m, n\}$. A more detailed recap of LoRA can be found in Apdx. A.1. Directional alignment can be achieved if singular vectors for $\Delta\mathbf{W}$ are leveraged to initialize $\mathbf{X}_0$ and $\mathbf{Y}_0$. While $\Delta\mathbf{W}$ is unavailable a priori, empirical wisdom suggests that there exists a set of well-performed adapters that lie in the column span of the pretrained weight matrix (Lingam et al., 2024), i.e., $\text{ColSpan}(\Delta\mathbf{W}) \subseteq \text{ColSpan}(\mathbf{W}_0)$. In other words, $\mathbf{W}_0$ can be adopted as a suitable replacement of $\Delta\mathbf{W}$ for directional alignment.

Having $\text{ColSpan}(\mathbf{W}_0)$ alone is insufficient for directional alignment, since it does not specify which directions are more crucial. To answer this question, we examine the singular values that undergo the most significant change after LoRA finetuning on a few-shot learning task (Malladi et al., 2023). OPT-1.3B is chosen as the base model and LoRA is applied to its query and value matrices with $r = 8$; more details can be found in Apdx. E.3. For each LoRA layer, we count the indices of $r$ singular values that exhibit the largest changes after finetuning, and summarize their frequencies across all layers in Fig. 2. It is observed that the top-$r$ singular values tend to have larger change, explaining the success of LoRA initialization approaches that aligns $\mathbf{X}_0$ with the directions corresponding to these singular values, such as PiSSA and OLoRA (Meng et al., 2024; Büyükakyüz, 2024). However, across all tested datasets, a substantial portion of non-top-$r$ singular values also demonstrate significant variation, and the frequency is positively linked to the singular values. In other words, the directions corresponding to larger singular values tend to be more important. This is akin to the principle of Nyström initialization $\mathbf{X}_0 = \mathbf{W}_0\mathbf{\Omega}$, evidenced by its spectrum, i.e., $\mathbb{E}[\mathbf{X}_0\mathbf{X}_0^\top] \propto \mathbf{W}_0\mathbf{W}_0^\top$.

Building upon these observations, and considering the accelerated convergence with Nyström initialization in ScaledGD, we propose two novel variants of LoRA:

- **Nyström LoRA (NoRA)**: This approach applies (5) directly on top of LoRA, that is, $\mathbf{X}_0 = \mathbf{W}_0\mathbf{\Omega}$ and $\mathbf{Y}_0 = \mathbf{0}$.
- **Nyström preconditioned LoRA (NoRA+)**: This approach not only advances LoRA initialization with (5), but also leverages ScaledGD for optimization.

We note that ScaledGD has already been applied for LoRA training in (Zhang & Pilanci, 2024), which we refer to as LoRA-P (P for preconditioning). We will show that both LoRA and LoRA-P benefit significantly from Nyström initialization. Due to space limitation, we summarize NoRA and NoRA+ in Algs. 1 and 2, respectively in the appendix, with additional explanations in Apdx. A.3.

**Deployment efficiency.** NoRA offers practical advantages over other initialization methods such as PiSSA and OLoRA. It not only bypasses the computationally expensive SVD or QR decomposition, but also avoids the need to modify to the pretrained weights. NoRA is thus an off-the-shelf solution to enhance LoRA without altering existing pipelines. We expand on this in Apdx. A.3.

Table 3: Training loss of NoRA and NoRA+ with stable-diffusion.

| loss($\downarrow$) | LoRA | LoRA-P | NoRA | NoRA+ |
|---|---|---|---|---|
| avg | 0.092±0.012 | 0.093±0.012 | 0.084±0.017 | 0.084±0.015 |

Figure 3: Generated images from NoRA and NoRA+ with stable-diffusion.

## 5 NUMERICAL RESULTS FOR NORA

The efficiency of proposed NoRA and NoRA+ is demonstrated on large-scale finetuning tasks involving diffusion and LLMs. The experiments are conducted with PyTorch (Paszke et al., 2019) on NVIDIA H100 GPUs. Details on datasets and experimental procedures can be found in Apdx. E. Code is available at https://github.com/BingcongLi/NoRA.

### 5.1 FEW-SHOT LEARNING WITH OPT-1.3B

Our evaluation starts with a few-shot learning task following (Malladi et al., 2023). The objective is to rapidly adapt a language model with a small training set. The datasets for this experiment are drawn from GLUE and SuperGLUE benchmarks (Wang et al., 2019b;a). Consistent with (Malladi et al., 2023), we randomly sample 1,000 data points for training and another 1,000 for testing.

We embrace OPT-1.3B as our base model (Zhang et al., 2022) and apply LoRA to the query and value matrices in the attention module. This aligns with common practice for models of this size. The rank of LoRA is set to 8, leading to approximately 1.5M trainable parameters, which is significantly less than the model size. We compare the proposed NoRA and NoRA+ with LoRA, prefix tuning (Li & Liang, 2021), OLoRA (Büyükakyüz, 2024), and PiSSA (Meng et al., 2024). Note that the latter two serve as alternative methods for initializing LoRA.

The performance of different algorithms is summarized in Tab. 2. It is evident that OLoRA, PiSSA, NoRA, and NoRA+ all outperform LoRA because their initialization strategies have provided more favorable directions for optimization. Among these initialization approaches, NoRA and NoRA+ have the best average accuracy, with absolute improvement over LoRA by 1.8 and 1.9, respectively.

### 5.2 SUBJECT-DRIVEN IMAGE GENERATION WITH STABLE-DIFFUSION

Next, we focus on subject-driven image generation (Ruiz et al., 2023). The goal of this task is to finetune a diffusion model with only a few user-specific images (typically less than 10) so that the modal can generate the same object in various contexts. The base model is selected as StableDiffusion v1.4 (Rombach et al., 2022) (0.98B parameters in total). We adhere to the default setting and finetune the U-Net with LoRA. The rank of LoRA is set as 4, amounting to 0.8M trainable parameters. The diffusion model is finetuned on a user-specific training set containing pictures of a dog labeled "a photo of $V_{dog}$," with the aim to generate proper images under the prompt "a $V_{dog}$ eating nachos."

Table 4: Performance of various algorithms for commonsense reasoning on LLaMA-7B and LLaMA2-7B. The results marked with ‡ are taken from (Liu et al., 2024). HS and WG are abbreviations for HellaSwag and WinoGrande, respectively.

| | Alg. | BoolQ | PIQA | SIQA | HS | WG | ARC-e | ARC-c | OBQA | avg (↑) |
|---|---|---|---|---|---|---|---|---|---|---|
| LLaMA | LoRA | 66.42 | 80.03 | 77.84 | 82.88 | 81.85 | 79.92 | 63.40 | 77.20 | 76.19 |
| | LoRA-P | 68.96 | 80.95 | 77.43 | 81.54 | 80.27 | 78.83 | 64.16 | 79.20 | 76.41 |
| | **NoRA** | 68.20 | 80.79 | 78.40 | 85.09 | 80.27 | 79.17 | 62.80 | 78.80 | **76.69** |
| | **NoRA+** | 69.85 | 81.83 | 77.38 | 82.09 | 80.03 | 79.67 | 64.25 | 78.60 | **76.71** |
| LLaMA2 | LoRA‡ | 69.8 | 79.9 | 79.5 | 83.6 | 82.6 | 79.8 | 64.7 | 81.0 | 77.6 |
| | LoRA-P | 71.47 | 81.50 | 78.81 | 85.97 | 80.43 | 81.14 | 66.55 | 81.00 | 78.35 |
| | **NoRA** | 71.16 | 83.08 | 79.53 | 85.90 | 81.85 | 80.64 | 66.13 | 81.80 | 78.76 |
| | **NoRA+** | 70.52 | 81.94 | 79.07 | 87.66 | 82.24 | 82.70 | 67.06 | 80.20 | **78.92** |

To demonstrate the power of initialization, we compare NoRA and NoRA+ with LoRA and LoRA-P. The averaged training loss of considered approaches are summarized in Tab. 3. It can be seen that NoRA and NoRA+ have $9.6\%$ smaller training loss compared with LoRA and LoRA-P, demonstrating the benefits of directional alignment at initialization. The generated images are listed in Fig. 3. Some of images generated by LoRA are not natural. For instance, the third one does not have a nice expression for nachos, and the tenth is not vivid. For LoRA-P, the dog in the third image is also not natural. NoRA and NoRA+, on the other hand, both generate high-fidelity pictures. However, there is a floating plate in the 8th image of NoRA+, but ensuring diffusion models to follow physical laws goes beyond the scope of this work. Additional results are provided in Apdx. E.5, where we finetune on images of a cat toy. The generated images from NoRA and NoRA+ have more lively facial details compare to those not using Nyström initialization.

### 5.3 Commonsense Reasoning with LLaMA-7B and LLaMA2-7B

Our evaluation is further scaled to LLMs using LLaMA and LLaMA2-7B (Touvron et al., 2023a;b). We tackle commonsense reasoning tasks following the setup in (Hu et al., 2023). Training data are merged from 8 datasets listed in Tab. 4. The test sets remain separate for individual evaluation. These reasoning tasks are intended to push the model beyond pattern recognition, requiring commonsense and knowledge to make proper inferences. The rank of LoRA is chosen as 32.

The results on LLaMA-7B are summarized in the upper block of Tab. 4. It is observed that NoRA improves the average accuracy by 0.5 over LoRA, while NoRA+ also surpasses LoRA-P. These results underscore the significance of initialization for optimizing LoRA. The numerical results on LLaMA2-7B are presented in the lower block of Tab. 4. It is observed that LoRA is unstable, henceforth the results for LoRA are taken from (Liu et al., 2024). This instability is not observed in other approaches tested. In this experiment, the benefit of the Nyström initialization is particularly pronounced, as the absolute improvement is even greater compared to the results on LLaMA-7B.

**Additional numerical results.** The efficiency of NoRA and NoRA+ is further validated on Gemma-7B for math reasoning tasks. More details can be found in Apdx. E.7.

## 6 Concluding remarks

This work characterizes how initialization can crucially determine the convergence behavior of the same optimization algorithm on matrix factorization problems. We prove that Nyström initialization can significantly improve the complexity bounds of ScaledGD under a wide spectrum of settings; see details in Tab. 1. One of the key improvements is that Nyström initialization enables a quadratic convergence for exact- and over-parametrized problems, whereas small initialization only guarantees a linear rate on ScaledGD. This performance gap calls for more careful investigation into the role of initialization in optimization. Additionally, the proposed Nyström initialization offers practical merits when applied on finetuning with LoRA, delivering deployment flexibility and promising numerical performance on large-scale problems with LLMs and diffusion models.

## ACKNOWLEDGEMENTS

We thank anonymous reviewers for their suggestions. BL is supported by Swiss National Science Foundation (SNSF) Project Funding No. 200021-207343. LZ gratefully acknowledges funding by the Max Planck ETH Center for Learning Systems (CLS). AM is partially supported by the NSF AI Institute for Foundations of Machine Learning (IFML), the NSF CAREER Award CCF-2338846, and a Google Research Scholar Award. NH is supported by ETH research grant funded through ETH Zurich Foundations and SNSF Project Funding No. 200021-207343.

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

# Supplementary Document for
# "On the Crucial Role of Initialization for Matrix Factorization"

## Contents

## A    MISSING DETAILS

### A.1    MORE ON RELATED WORK

**Convergence of over-parametrized matrix factorization problems.** Consider again the asymmetric problem as an example, i.e., $\min_{\mathbf{X},\mathbf{Y}} \|\mathbf{X}\mathbf{Y}^\top - \mathbf{A}\|^2$ with $\mathbf{A} \in \mathbb{R}^{m \times n}$, $\mathbf{X} \in \mathbb{R}^{m \times r}$ and $\mathbf{Y} \in \mathbb{R}^{n \times r}$. Over-parametrization refers to the case where $\text{rank}(\mathbf{A}) \leq r$. The gradient flow on the extreme over-parametrized problems, where $r \geq \max\{m, n\}$, is studied in (Tarmoun et al., 2021). There are also papers (Stöger & Soltanolkotabi, 2021; Zhang et al., 2021; Xiong et al., 2024) considering the matrix sensing problem, which partially relates to our problem when there are sufficient Gaussian measures. The work of (Arora et al., 2018) considers deeper problem (i.e., having more than 3 layers) while assuming $\mathbf{A}$ is full rank. Our results on over-parametrization can be found in Apdx. B.4 and Apdx. C.3 for symmetric and asymmetric problems, respectively. The comparison of ScaledGD with other works on over-parametrized problems can be found in Tab. 1.

**Convergence of quasi-Newton methods.** ScaledGD is sometimes regarded as a quasi-Newton method (Tong et al., 2021) for our nonconvex objective (4). Since much of the existing work on this topic focuses on the smooth and strongly convex case, we briefly review these approaches to highlight the significance of our results in achieving quadratic convergence for nonconvex and local-smooth problems. In the smooth and strongly convex regime, the primary advantage of quasi-Newton methods lies in their asymptotic ability to achieve super-linear convergence as $t \to \infty$. Non-asymptotic analyses demonstrating super-linear local rates have only been established recently; see, e.g., (Rodomanov & Nesterov, 2021; Jin & Mokhtari, 2023; Ye et al., 2023; Jiang et al., 2023c).

**LoRA and parameter-efficient finetuning.** LoRA (Hu et al., 2022) is a notable example of parameter-efficient finetuning (PEFT) approaches. The goal of PEFT is to reduce the resource requirement for finetuning LLMs on downstream tasks. Other commonly adopted PEFT methods include, e.g., adapters (Houlsby et al., 2019) and prefix tuning (Li & Liang, 2021). There are also various efforts to further enhance LoRA via adaptivity (Zhang et al., 2023), chaining (Lialin et al., 2024; Xia et al., 2024), low-bit training (Dettmers et al., 2023; Li et al., 2024b), regularization (Li et al., 2024a), modifications for long-sequences (Chen et al., 2024), weight decomposition (Liu et al., 2024), and combining with sparsity (Nikdan et al., 2024). Additionally, there are several approaches aiming at further reducing the number of trainable parameters in LoRA; examples include (Kopiczko et al., 2024; Lingam et al., 2024; Gao et al., 2024; Zhu et al., 2024; Hao et al., 2024; Bałazy et al., 2024). While originally designed for finetuning LLMs, LoRA also finds its applications in other domains, such as image generation (Gu et al., 2023) and continual learning (Smith et al., 2023).

**LoRA initialization.** When first proposed, LoRA initialization was largely overlooked. The work of (Hayou et al., 2024) justifies that whether setting $\mathbf{X}_0$ or $\mathbf{Y}_0$ to be $\mathbf{0}$ affects performance from a stability perspective. Recent works (Büyükakyüz, 2024; Meng et al., 2024) observe a fundamental difference between initialization of LoRA and neural networks, emphasizing the availability of prior knowledge. These works experimentally demonstrate that pretrained model can serve as prior to guide the direction of adapters, and hence perform QR or SVD on the pretrained matrix and using (scaled) top-$r$ singular vectors for LoRA initialization. Follow-up study (Wang et al., 2024) exploits stability for further improvement. However, these initialization methods are computationally expensive and lack flexibility for deployment. The proposed NoRA initialization overcomes these limitations.

### A.2    LoRA FOR LINEAR MODELS AS ASYMMETRIC MATRIX FACTORIZATION

We argue that LoRA applied on linear models given a whitened dataset is equivalent to the asymmetric matrix factorization problem. The whitened dataset is widely adopted for theoretical analyses, and we refer to (Arora et al., 2018; Jiang et al., 2023a; Yaras et al., 2024) for more details.

Assume that we have a pretrained (linear) model $\mathbf{W}_0 \in \mathbb{R}^{m \times n}$. Applying LoRA on this layer with whitened data $\mathbf{B}$ is equivalent to solving the following problem

$$\frac{1}{2}\|(\mathbf{W}_0 + \mathbf{X}\mathbf{Y}^\top) - \mathbf{B}\|_\mathsf{F}^2. \tag{8}$$

It is clearly that this problem (8) is the same as (4) by setting $\mathbf{A} = \mathbf{B} - \mathbf{W}_0$.

Unfortunately, existing works provide no theoretical support on the most widely adopted initialization approach for LoRA in practice – either $\mathbf{X}_0$ or $\mathbf{Y}_0$ is chosen as $\mathbf{0}$ to preserve $\mathbf{W}_0 + \mathbf{X}_0\mathbf{Y}_0^\top = \mathbf{W}_0$.

| **Algorithm 1** NoRA for a specific LoRA layer | **Algorithm 2** NoRA+ for a specific LoRA layer |
|---|---|
| 1: **Initialize:** $\xi$ – standard deviation of random matrix $\boldsymbol{\Omega}$ 
 2: Set $\mathbf{X}_0$ and $\mathbf{Y}_0$ via Nyström initialization (5) 
 3: Standard training process | 1: **Initialize:** $\xi$ – standard deviation of random matrix $\boldsymbol{\Omega}$; $\lambda$ – numerical stability of matrix inversion 
 2: Set $\mathbf{X}_0$ and $\mathbf{Y}_0$ via Nyström initialization (5) 
 3: **for** $t = 0, \ldots, T-1$ **do** 
 4:      Get gradient $\mathbf{G}_{\mathbf{X}_t}$ and $\mathbf{G}_{\mathbf{Y}_t}$ 
 5:      **if** $t > 0$ **then** 
 6:          $\mathbf{G}_{\mathbf{X}_t} \leftarrow \mathbf{G}_{\mathbf{X}_t}(\mathbf{Y}_t^\top\mathbf{Y}_t + \lambda\mathbf{I}_r)^{-1}/\|(\mathbf{Y}_t^\top\mathbf{Y}_t + \lambda\mathbf{I}_r)^{-1}\|_{\mathsf{F}}$ 
 7:      **end if** 
 8:      $\mathbf{G}_{\mathbf{Y}_t} \leftarrow \mathbf{G}_{\mathbf{Y}_t}(\mathbf{X}_t^\top\mathbf{X}_t + \lambda\mathbf{I}_r)^{-1}/\|(\mathbf{X}_t^\top\mathbf{X}_t + \lambda\mathbf{I}_r)^{-1}\|_{\mathsf{F}}$ 
 9:      Optimizer update 
 10: **end for** |

In this sense, our Nyström initialization in (5) is the first means of initialization that justifies one variable can be set to $\mathbf{0}$.

**Additional similarities between LoRA and matrix factorization.** LoRA and matrix factorization share similar mathematical properties. For example, they both have no spurious local minima (Du et al., 2018; Ge et al., 2017; Jang et al., 2024). There are also recent efforts using insights from matrix factorization to further improve LoRA; see e.g., (Yaras et al., 2024; Nikdan et al., 2024).

## A.3 MORE ON NORA AND NORA+

As discussed in Sec. 4, LoRA can significantly benefit from the aligned directions at initialization. Besides the theoretical benefits of applying Nyström initialization on ScaledGD (NoRA+), Nyström initialization can also be used directly with Adam (or AdamW), i.e., NoRA. There are several reasons for this. First, directional alignment from initialization is beneficial to most optimizers. While our theoretical results focus on ScaledGD, we believe that the aligned directions also improve GD. Despite the improvement may be less significant as in ScaledGD, we conjecture that the linear term in (Ye & Du, 2021, Theorem 1.1) can be removed with Nyström initialization, because it can be roughly understood as the price of searching for proper directions. In other words, the benefits of Nyström initialization extend to other optimizers as well. Second, Adam also affords an explanation of preconditioning, and the preconditioner for $\mathbf{X}_t$ is also closely related to $\mathbf{Y}_t$. In other words, Adam shares similarities with ScaledGD in (6). These two reasons prompt the proposed NoRA, as summarized in Alg. 1. For NoRA+ in Alg. 2, we modify the vanilla ScaledGD iterations in (6) with two add-ons. First, a small parameter $\lambda$ is introduced for numerical stability of matrix inversion. This is a standard practice for numerical optimizers such as Adam (Kingma & Ba, 2014; Loshchilov & Hutter, 2017). Second, the gradient is normalized by the Frobenius norm of its preconditioner. The reason is that an optimal $\lambda$ is difficult to tune as shown in (Zhang & Pilanci, 2024), where they use $\lambda$ from $10^{-6}$ to 100. With this normalizer, we can set $\lambda = 10^{-6}$ in all our experiments without any tuning. Moreover, this normalizer is useful to prevent the instability in earlier iterations due to the non-invertable $\mathbf{Y}_0 = \mathbf{0}$.

**Deployment efficiency of NoRA.** One benefit of NoRA (as well as NoRA+) is that it can be deployed jointly with adapters trained with LoRA – and hence there is no need to modify the current pipeline for deployment. This is because both of NoRA and LoRA do not need to modify the pretrained parameters, and the finetuned model is just $\mathbf{W}_0 + \mathbf{X}_T\mathbf{Y}_T^\top$, where $\mathbf{W}_0$ is the pretrained model, and $\mathbf{X}_T$ and $\mathbf{Y}_T$ are finetuned adapter weights. On the contrary, other initialization approaches such as PiSSA and OLoRA (Meng et al., 2024; Büyükakyüz, 2024) are less efficient for using jointly with LoRA at deployment because both approaches modify the pretrained weights, so that the finetuned model becomes $\widehat{\mathbf{W}}_0 + \mathbf{X}_T\mathbf{Y}_T^\top$, where $\widehat{\mathbf{W}}_0 = \mathbf{W}_0 - \mathbf{X}_0\mathbf{Y}_0^\top$. The use of $\widehat{\mathbf{W}}_0$ comes from the fact that initialization in PiSSA and OLoRA does not satisfy $\mathbf{X}_0\mathbf{Y}_0^\top = \mathbf{0}$. Consequently, when deploying PiSSA jointly with LoRA, one needs to store both $\mathbf{W}_0$ (for LoRA) and $\widehat{\mathbf{W}}_0$ (for PiSSA), leading to reduced memory efficiency.

### A.4  INITIALIZATION WITHOUT KNOWING $\mathbf{A}$

In certain scenarios, direct access to $\mathbf{A}$ in problems (1) and (4) may not be available. Nevertheless, the Nyström initialization remains applicable, as it can be derived from the gradient, which is the minimum requirement for gradient-based methods. We take exact-parametrization as an illustrative example. For the symmetric problem (1), Nyström initialization (3) can be obtained from the gradient at $\mathbf{\Omega}$, i.e., $\mathbf{G}_0 = (\mathbf{\Omega}\mathbf{\Omega}^\top - \mathbf{A})\mathbf{\Omega}$ and set initialization as $\mathbf{X}_0 = -\mathbf{G}_0 + \mathbf{\Omega}\mathbf{\Omega}^\top\mathbf{\Omega}$. Similarly, for the asymmetric problem (4), our Nyström initialization in (5) can be obtained via the negative gradient at $\mathbf{X} = 0$ and $\mathbf{Y} = \mathbf{\Omega}$.

### A.5  SCALEDGD AS A NEW MATRIX FACTORIZATION FOR ASYMMETRIC PROBLEMS

The one-step convergence of ScaledGD for asymmetric matrix factorization (4) can be interpreted as a new method to factorize matrix $\mathbf{A}$. Taking the exact-parametrized case as an example, this new factorization induced with ScaledGD in (6) has several computational benefits.

❖ **Lower computational complexity than SVD in certain regime.** Observing that the computation for (6) mostly arises from matrix multiplication, whose complexity can be reduced with state-of-the-art algorithms (Williams et al., 2024; Alman et al., 2024). In particular, multiplication of two square matrix of size $n \times n$ can be done with complexity $\mathcal{O}(n^{2.38})$. To apply this on ScaledGD in (6), we can pad the rectangular matrices with $0$ to make them square. Simple calculation shows that when $r > \mathcal{O}(n^{0.38})$, the complexity is lower than truncated SVD. Additionally, our derivation does not account for the sparsity of the padded matrices, suggesting that the theoretical computational complexity could be further reduced.

❖ **Potential acceleration with GPUs.** Note that the computational pattern of (6) enables it to embrace hardware acceleration provided by e.g., GPUs. In particular, (6) extensively uses matrix multiplication, and there is no vector operation. This is in stark contrast to SVD, which needs sequential householder transformation. Such sequential operations of SVD is often performed on CUDA cores, while (6) can be done in Tensor cores (suitable for matrix multiplication). This enables (6) to enjoy the massive benefits of Tensor core, such as computation with mixed precision, and significant throughput improvement (roughly 15x on H100).[3]

### A.6  ADDITIONAL DISCUSSIONS

**An alternative interpretation of our results.** Our theoretical results can also be interpreted as emphasizing the importance of identifying the correct directions when optimizing functions with fourth-order growth. In the matrix factorization and sensing literature, it is common for algorithms such as GD or ScaledGD to exhibit a two-phase behavior: an initial phase of selecting the correct direction (also known as the spectral phase), followed by a second phase of rapid (e.g., linear) local convergence. Our findings clearly indicate that the correct directions in the first phase are critical – not only for faster termination of this phase, more importantly, for exponentially impacting the convergence rate in the second phase of ScaledGD.

**Future directions.** Our findings present several avenues for further exploration. From a theoretical standpoint, this work focuses on the impact of initialization in canonical matrix factorization problems. We believe our results can be extended to more complex settings, such as matrix sensing and tensor factorization, which are part of our future research plans. On the practical side, our work hints at the potential for further gains by leveraging priors embedded in pretrained weights within the context of Burer-Monteiro factorization with LoRA. Investigating how to better uncover and utilize this hidden information is another attractive direction for future research.

---

[3]https://www.ece.lsu.edu/koppel/gp/refs/gtc22-whitepaper-hopper.pdf

## B  Missing proofs for symmetric settings

### B.1  Initialization of exact- and under-parametrized problems

#### B.1.1  Proof of Lemma 1

*Proof.* Let the compact eigenvalue decomposition of $\mathbf{A}$ be $\mathbf{A} = \mathbf{Q}\boldsymbol{\Sigma}\mathbf{Q}^\top$, where $\mathbf{Q} \in \mathbb{R}^{m \times r_A}$ and $\boldsymbol{\Sigma} \in \mathbb{R}^{r_A \times r_A}$. We then have that

$$\mathbf{X}_0 = (\mathbf{Q}\boldsymbol{\Sigma})(\mathbf{Q}^\top\boldsymbol{\Omega}). \tag{9}$$

It is not hard to verify that the matrix $\mathbf{Q}^\top\boldsymbol{\Omega} \in \mathbb{R}^{r_A \times r}$ is also a Gaussian random matrix, where each entry follows $\mathcal{N}(0, \xi^2)$. Applying Lemma 19 on $\mathbf{Q}^\top\boldsymbol{\Omega}$, it can be seen that

$$\mathbb{P}\Big(\frac{\sigma_r(\mathbf{Q}^\top\boldsymbol{\Omega})}{\xi} \le \tau(\sqrt{r_A} - \sqrt{r-1})\Big) \le (C_1\tau)^{r_A - r + 1} + e^{-C_2 r_A} := \delta$$

where $C_1$ and $C_2$ are universal constants independent of $r_A$ and $r$. This inequality shows that with probability at least $1 - \delta$, $\sigma_r(\mathbf{Q}^\top\boldsymbol{\Omega}) \ge \xi\tau(\sqrt{r_A} - \sqrt{r-1})$.

Note that inequality $\sigma_{\min}(\mathbf{CD}) \ge \sigma_{\min}(\mathbf{C})\sigma_{\min}(\mathbf{D})$ holds given full column rank of $\mathbf{C}$; see Lemma 17. Applying it to (9), we have that

$$\sigma_r(\mathbf{X}_0) \ge \sigma_{r_A}(\mathbf{Q}\boldsymbol{\Sigma})\sigma_r(\mathbf{Q}^\top\boldsymbol{\Omega}) = \sigma_{r_A}(\mathbf{A})\sigma_r(\mathbf{Q}^\top\boldsymbol{\Omega})$$
$$\overset{(a)}{\ge} \xi\tau(\sqrt{r_A} - \sqrt{r-1})\sigma_{r_A}(\mathbf{A})$$

where (a) holds with probability at least $1 - \delta$. $\qquad\square$

### B.2  Missing proofs for the symmetric and exact-parametrized setting

In the exact-parametrized setting, it is convenient to define

$$\mathbf{B}_t := \boldsymbol{\Phi}_t\boldsymbol{\Phi}_t^\top \tag{10}$$

where $\boldsymbol{\Phi}_t \in \mathbb{R}^{r \times r}$ comes from Lemma 2, i.e., $\mathbf{X}_t = \mathbf{Q}\boldsymbol{\Phi}_t$. The notation $\mathbf{B}_t$ will be used frequently in this subsection. With the help of Lemma 2, $\mathbf{B}_t$ can be understood as the "core" part of $\mathbf{X}_t\mathbf{X}_t^\top$, because $\mathbf{X}_t\mathbf{X}_t^\top = \mathbf{Q}\boldsymbol{\Phi}_t\boldsymbol{\Phi}_t^\top\mathbf{Q}^\top = \mathbf{Q}\mathbf{B}_t\mathbf{Q}^\top$. Once proving Lemma 2, it allows us to study dynamics using a simpler but equivalent notion $\|\mathbf{B}_t - \boldsymbol{\Sigma}\|_{\mathsf{F}}$, i.e.,

$$\|\mathbf{X}_t\mathbf{X}_t^\top - \mathbf{A}\|_{\mathsf{F}} = \|\mathbf{Q}(\boldsymbol{\Phi}_t\boldsymbol{\Phi}_t^\top - \boldsymbol{\Sigma})\mathbf{Q}^\top\|_{\mathsf{F}} = \|\boldsymbol{\Phi}_t\boldsymbol{\Phi}_t^\top - \boldsymbol{\Sigma}\|_{\mathsf{F}} = \|\mathbf{B}_t - \boldsymbol{\Sigma}\|_{\mathsf{F}}.$$

#### B.2.1  Proof of Lemma 2

*Proof.* The proof relies on $\mathbf{B}_t$ defined in (10). We will prove this lemma by induction. Since $\mathbf{X}_0 = \mathbf{A}\boldsymbol{\Omega}$ in Nyström initialization, we have that $\boldsymbol{\Phi}_0 = \boldsymbol{\Sigma}\mathbf{Q}^\top\boldsymbol{\Omega}$. Moreover, our base assumption $\sigma_r(\mathbf{B}_0) > 0$ is true because $\text{rank}(\mathbf{B}_0) = \text{rank}(\mathbf{X}_0\mathbf{X}_0^\top) = r$, which is the result of Lemma 1.

For induction, assume that $\mathbf{X}_t$ can be written as $\mathbf{X}_t = \mathbf{Q}\boldsymbol{\Phi}_t$ with a full rank $\boldsymbol{\Phi}_t \in \mathbb{R}^{r \times r}$ at iteration $t$. By the update (2), we have that

$$\begin{aligned}
\mathbf{X}_{t+1} &= \mathbf{X}_t - \eta(\mathbf{X}_t\mathbf{X}_t^\top - \mathbf{A})\mathbf{X}_t(\mathbf{X}_t^\top\mathbf{X}_t)^{-1} \\
&= \mathbf{Q}\boldsymbol{\Phi}_t - \eta\mathbf{Q}(\boldsymbol{\Phi}_t\boldsymbol{\Phi}_t^\top - \boldsymbol{\Sigma})\mathbf{Q}^\top\mathbf{Q}\boldsymbol{\Phi}_t(\boldsymbol{\Phi}_t^\top\mathbf{Q}^\top\mathbf{Q}\boldsymbol{\Phi}_t)^{-1} \\
&\overset{(a)}{=} \mathbf{Q}\Big(\boldsymbol{\Phi}_t - \eta(\boldsymbol{\Phi}_t\boldsymbol{\Phi}_t^\top - \boldsymbol{\Sigma})\boldsymbol{\Phi}_t(\boldsymbol{\Phi}_t^\top\boldsymbol{\Phi}_t)^{-1}\Big) \\
&\overset{(b)}{=} \mathbf{Q}\underbrace{\Big((1 - \eta)\boldsymbol{\Phi}_t + \eta\boldsymbol{\Sigma}\boldsymbol{\Phi}_t^{-\top}\Big)}_{:= \boldsymbol{\Phi}_{t+1}},
\end{aligned} \tag{11}$$

where (a) uses $\mathbf{Q}^\top\mathbf{Q} = \mathbf{I}_r$; and (b) uses $\boldsymbol{\Phi}_t$ is full rank (hence invertible). Note that $\mathbf{Q}$ and $\mathbf{A}$ share the same column space. This proves the first claim i) of this lemma.

Next we show that the smallest eigenvalue of $\mathbf{B}_{t+1}$ is bounded away from 0, or equivalently, $\mathbf{\Phi}_{t+1}$ is full rank. To start with, we have that from the expression of $\mathbf{\Phi}_{t+1}$ in (11),

$$
\begin{aligned}
\mathbf{B}_{t+1} = \mathbf{\Phi}_{t+1}\mathbf{\Phi}_{t+1}^\top &= (1-\eta)^2\mathbf{\Phi}_t\mathbf{\Phi}_t^\top + 2\eta(1-\eta)\mathbf{\Sigma} + \eta^2\mathbf{\Sigma}\mathbf{\Phi}_t^{-\top}\mathbf{\Phi}_t^{-1}\mathbf{\Sigma} \\
&= (1-\eta)^2\mathbf{B}_t + 2\eta(1-\eta)\mathbf{\Sigma} + \eta^2\mathbf{\Sigma}\mathbf{B}_t^{-1}\mathbf{\Sigma}.
\end{aligned}
\tag{12}
$$

Note that $\mathbf{B}_{t+1}$ is a PSD matrix by definition (hence the eigenvalues and singular values are the same). To see the smallest eigenvalue of $\mathbf{B}_{t+1}$ is lower bounded, we will apply Lemma 15 on (12) twice, i.e.,

$$
\begin{aligned}
&\sigma_r(\mathbf{B}_{t+1}) \\
&\overset{(c)}{\geq} 2\eta(1-\eta)\sigma_r(\mathbf{\Sigma}) + \sigma_r\Big((1-\eta)^2\mathbf{B}_t + \eta^2\mathbf{\Sigma}\mathbf{B}_t^{-1}\mathbf{\Sigma}\Big) \\
&\overset{(d)}{\geq} 2\eta(1-\eta)\sigma_r(\mathbf{\Sigma}) + (1-\eta)^2\sigma_r(\mathbf{B}_t) \\
&\overset{(e)}{\geq} (1-\eta)^{2t+2}\sigma_r(\mathbf{B}_0) + 2\eta(1-\eta)\sigma_r(\mathbf{\Sigma})\frac{1-(1-\eta)^{2t+2}}{2\eta-\eta^2} \\
&\overset{(f)}{\geq} (1-\eta)^{2t+2}\sigma_r(\mathbf{B}_0) + (1-\eta)\sigma_r(\mathbf{\Sigma}) - (1-\eta)^{2t+3}\sigma_r(\mathbf{\Sigma}),
\end{aligned}
\tag{13}
$$

where (c) and (d) are because of Lemma 15; (e) is by unrolling $\sigma_r(\mathbf{B}_t)$ using (d); and (f) is by $\frac{2\eta}{2\eta-\eta^2}\geq 1$. Combining (11) and (13) concludes the induction. $\qquad\square$

### B.2.2 PROOF OF THEOREM 1

*Proof.* The proof is by combining Lemmas 7 and 8. $\qquad\square$

**Lemma 7** (Phase I. Linear convergence to near optima). *Let* $\eta = \mathcal{O}(\frac{1}{\kappa^3\|\mathbf{A}\|_\mathsf{F}})$. *After* $\mathcal{O}(\kappa^3\sqrt{r}\log\kappa)$ *iterations, ScaledGD* (2) *with Nyström initialization* (3) *ensures that* $\|\mathbf{X}_t\mathbf{X}_t^\top - \mathbf{A}\|_\mathsf{F} \leq \mathcal{O}(1/\kappa^2)$.

*Proof.* Subtracting $\mathbf{\Sigma}$ from both sides of (12), we can obtain that
$$
\mathbf{B}_{t+1} - \mathbf{\Sigma} = (1-\eta)^2(\mathbf{B}_t - \mathbf{\Sigma}) - \eta^2\mathbf{\Sigma} + \eta^2\mathbf{\Sigma}\mathbf{B}_t^{-1}\mathbf{\Sigma}.
$$
This implies that

$$
\begin{aligned}
&\|\mathbf{B}_{t+1} - \mathbf{\Sigma}\|_\mathsf{F} \\
&\overset{(a)}{\leq} (1-\eta)^2\|\mathbf{B}_t - \mathbf{\Sigma}\|_\mathsf{F} + \eta^2\|\mathbf{\Sigma}\|_\mathsf{F} + \eta^2\|\mathbf{\Sigma}\mathbf{B}_t^{-1}\|_2\|\mathbf{\Sigma}\|_\mathsf{F} \\
&\overset{(b)}{\leq} (1-\eta)^2\|\mathbf{B}_t - \mathbf{\Sigma}\|_\mathsf{F} + \eta^2\|\mathbf{\Sigma}\|_\mathsf{F} + \eta^2\|\mathbf{\Sigma}\|_2\|\mathbf{B}_t^{-1}\|_2\|\mathbf{\Sigma}\|_\mathsf{F} \\
&\leq (1-\eta)\|\mathbf{B}_t - \mathbf{\Sigma}\|_\mathsf{F} + \eta^2\|\mathbf{\Sigma}\|_\mathsf{F} + \eta^2\frac{\sigma_1(\mathbf{\Sigma})\|\mathbf{\Sigma}\|_\mathsf{F}}{\sigma_r(\mathbf{B}_t)}
\end{aligned}
$$

where (a) is by $\|\mathbf{M}\mathbf{N}\|_\mathsf{F} \leq \|\mathbf{M}\|_2\|\mathbf{N}\|_\mathsf{F}$; and (b) follows from the sub-multiplicity of $\|\cdot\|_2$.

By Lemma 2, if $\eta \leq 2/3$ and there exists $T_1$ such that $\sigma_r(\mathbf{B}_{T_1}) \geq \sigma_r(\mathbf{\Sigma})/3$, then it holds that $\sigma_r(\mathbf{B}_t) \geq \sigma_r(\mathbf{\Sigma})/3, \forall t \geq T_1$. According to Lemma 1, we can choose $\xi$ in (3) sufficiently large such that $\sigma_r(\mathbf{B}_0) \geq \sigma_r(\mathbf{\Sigma})/3$, i.e., $T_1 = 0$. Alternatively, to avoid such a requirement on $\xi$, we can simply choose a constant step size, e.g., $\eta = 0.5$, and run a constant number of steps, $T_1 = \mathcal{O}(1/\eta)$, to ensure $\sigma_r(\mathbf{B}_{T_1}) \geq \sigma_r(\mathbf{\Sigma})/3$; see Lemma 2. For simplicity of the results, our proof below goes with the first method, i.e., $T_1 = 0$.

$$
\begin{aligned}
&\|\mathbf{B}_{t+1} - \mathbf{\Sigma}\|_\mathsf{F} \\
&\leq (1-\eta)\|\mathbf{B}_t - \mathbf{\Sigma}\|_\mathsf{F} + \eta^2\|\mathbf{\Sigma}\|_\mathsf{F} + \eta^2\frac{\sigma_1(\mathbf{\Sigma})\|\mathbf{\Sigma}\|_\mathsf{F}}{\sigma_r(\mathbf{B}_t)} \\
&\leq (1-\eta)\|\mathbf{B}_t - \mathbf{\Sigma}\|_\mathsf{F} + \eta^2\|\mathbf{\Sigma}\|_\mathsf{F} + 3\eta^2\frac{\sigma_1(\mathbf{\Sigma})\|\mathbf{\Sigma}\|_\mathsf{F}}{\sigma_r(\mathbf{\Sigma})} \\
&\overset{(c)}{\leq} \eta\|\mathbf{\Sigma}\|_\mathsf{F} + 3\eta\kappa\|\mathbf{\Sigma}\|_\mathsf{F} + (1-\eta)^{t+1-T_1}\|\mathbf{B}_{T_1} - \mathbf{\Sigma}\|_\mathsf{F} \\
&= \eta\|\mathbf{A}\|_\mathsf{F} + 3\eta\kappa\|\mathbf{A}\|_\mathsf{F} + (1-\eta)^{t+1-T_1}\|\mathbf{B}_{T_1} - \mathbf{\Sigma}\|_\mathsf{F}
\end{aligned}
$$

where (c) is by Lemma 14. From this inequality it is not difficult to see that once $\eta = \mathcal{O}(\frac{1}{\kappa^3 \|\mathbf{A}\|_F})$, one will have $\|\mathbf{B}_{t+1} - \boldsymbol{\Sigma}\|_F \leq \mathcal{O}(1/\kappa^2)$ within the stated iterations. $\square$

**Lemma 8** (Phase II. Quadratic convergence to global optima). *If we choose $\eta = 0.5$ and suppose that after $T_2$ iterations, $\sigma_r(\mathbf{B}_{T_2}) \geq \sigma_r(\boldsymbol{\Sigma})/3$ and $\|\mathbf{B}_{T_2} - \boldsymbol{\Sigma}\|_F \leq 2/(3\kappa^2)$ are satisfied, ScaledGD then ensures that for any $t \geq T_2$,*

$$\|\mathbf{X}_{t+1}\mathbf{X}_{t+1}^\top - \mathbf{A}\|_F = \|\mathbf{B}_{t+1} - \boldsymbol{\Sigma}_r\|_F \leq \frac{4}{3\kappa^2}\frac{1}{2^{2^{t+1}}}.$$

*Proof.* Let $\mathbf{C}_t = \boldsymbol{\Sigma}^{-1}\mathbf{B}_t$. We can rewrite (12) as

$$\mathbf{C}_{t+1} = (1-\eta)^2\mathbf{C}_t + 2\eta(1-\eta)\mathbf{I}_r + \eta^2\mathbf{C}_t^{-1}.$$

Subtracting $\mathbf{I}_r$ and rearranging it, we arrive at

$$\mathbf{C}_{t+1} - \mathbf{I}_r = (1-2\eta)(\mathbf{C}_t - \mathbf{I}_r) + \eta^2\mathbf{C}_t^{-1}(\mathbf{C}_t - \mathbf{I}_r)^2.$$

By choosing $\eta = 0.5$, we have that

$$\mathbf{C}_{t+1} - \mathbf{I}_r = \frac{1}{4}\mathbf{C}_t^{-1}(\mathbf{C}_t - \mathbf{I}_r)^2.$$

Multiplying both sides with $\boldsymbol{\Sigma}$, we have that

$$\mathbf{B}_{t+1} - \boldsymbol{\Sigma} = \frac{1}{4}\boldsymbol{\Sigma}\mathbf{B}_t^{-1}\boldsymbol{\Sigma}(\mathbf{C}_t - \mathbf{I}_r)(\mathbf{C}_t - \mathbf{I}_r)$$

$$= \frac{1}{4}\boldsymbol{\Sigma}\mathbf{B}_t^{-1}(\mathbf{B}_t - \boldsymbol{\Sigma})\boldsymbol{\Sigma}^{-1}(\mathbf{B}_t - \boldsymbol{\Sigma}).$$

This implies that

$$\|\mathbf{B}_{t+1} - \boldsymbol{\Sigma}\|_F \leq \frac{1}{4}\|\boldsymbol{\Sigma}\|_2\|\mathbf{B}_t^{-1}\|_2\|\mathbf{B}_t - \boldsymbol{\Sigma}\|_F\|\boldsymbol{\Sigma}^{-1}\|_2\|\mathbf{B}_t - \boldsymbol{\Sigma}\|_F$$

$$\overset{(a)}{\leq} \frac{3}{4}\frac{\sigma_1(\boldsymbol{\Sigma})}{\sigma_r^2(\boldsymbol{\Sigma})}\|\mathbf{B}_t - \boldsymbol{\Sigma}\|_F^2 \overset{(b)}{=} \frac{3\kappa^2}{4}\|\mathbf{B}_t - \boldsymbol{\Sigma}\|_F^2$$

where (a) is by Lemma 2, i.e., once $\sigma_r(\mathbf{B}_{T_2}) \geq \sigma_r(\boldsymbol{\Sigma})/3$, then $\sigma_r(\mathbf{B}_t) \geq \sigma_r(\boldsymbol{\Sigma})/3$ holds for all $t \geq T_2$; and (b) is by $\sigma_1(\boldsymbol{\Sigma}) = 1$ and $\sigma_r(\boldsymbol{\Sigma}) = 1/\kappa$.

Finally, applying Lemma 16, it can be seen that a quadratic rate can be established long as $\|\mathbf{B}_{T_2} - \boldsymbol{\Sigma}\|_F \leq \frac{2}{3\kappa^2}$, and this condition is satisfied from Lemma 7. $\square$

### B.3 MISSING PROOFS FOR THE SYMMETRIC AND UNDER-PARAMETRIZED SETTING

We start with some notation that would be helpful for this subsection. Let the compact eigenvalue decomposition of $\mathbf{A} = \mathbf{Q}\boldsymbol{\Sigma}\mathbf{Q}^\top$, where $\mathbf{Q} \in \mathbb{R}^{m \times r_A}$, and $\boldsymbol{\Sigma} \in \mathbb{R}^{r_A \times r_A}$.

In Lemma 4, we will prove that $\mathbf{X}_t = \mathbf{Q}\boldsymbol{\Phi}_t$ always holds if we employ Nyström initialization and ScaledGD in (2), where $\boldsymbol{\Phi}_t \in \mathbb{R}^{r_A \times r}$. We also denote $\boldsymbol{\Theta}_t := \boldsymbol{\Phi}_t(\boldsymbol{\Phi}_t^\top\boldsymbol{\Phi}_t)^{-1}$, where the invertibility of $(\boldsymbol{\Phi}_t^\top\boldsymbol{\Phi}_t)$ will become clear in the proof.

Lastly, let $\mathbf{B}_t := \boldsymbol{\Phi}_t^\top\boldsymbol{\Sigma}^{-1}\boldsymbol{\Phi}_t$. Note that $\mathbf{B}_t \in \mathbb{R}^{r \times r}$ and $\mathbf{B}_t = \mathbf{X}_t^\top\mathbf{A}^\dagger\mathbf{X}_t$.

#### B.3.1 PROOF OF LEMMA 3

*Proof.* We start with rewriting $\mathbf{A}$,

$$\mathbf{A} = [\mathbf{Q}_1, \mathbf{Q}_2]\begin{bmatrix}\boldsymbol{\Sigma}_1 & \mathbf{0} \\ \mathbf{0} & \boldsymbol{\Sigma}_2\end{bmatrix}\begin{bmatrix}\mathbf{Q}_1^\top \\ \mathbf{Q}_2^\top\end{bmatrix} = \mathbf{Q}_1\boldsymbol{\Sigma}_1\mathbf{Q}_1^\top + \mathbf{Q}_2\boldsymbol{\Sigma}_2\mathbf{Q}_2^\top \quad (14)$$

where $\mathbf{Q}_1 \in \mathbb{R}^{m \times r}$ and $\mathbf{Q}_2 \in \mathbb{R}^{m \times (r_A - r)}$ are the first $r$ and other columns of $\mathbf{Q}$, respectively; and $\boldsymbol{\Sigma}_1 \in \mathbb{R}^{r \times r}$ and $\boldsymbol{\Sigma}_2 \in \mathbb{R}^{(r - r_A) \times (r - r_A)}$ are diagonal matrices formed by the first $r$ and the rest diagonal entries of $\boldsymbol{\Sigma}$.

It is not difficult to see that the optimal solution of (1) is $\mathbf{X}_* = \mathbf{Q}_1 \boldsymbol{\Sigma}_1^{1/2} \mathbf{U}^\top$, where $\mathbf{U} \in \mathbb{R}^{r \times r}$ is any unitary matrix that accounts for rotation. Note that the pseudo-inverse of $\mathbf{A}$ can be written as $\mathbf{A}^\dagger = \mathbf{Q} \boldsymbol{\Sigma}^{-1} \mathbf{Q}^\top$. Plugging $\mathbf{X}_*$ into the definition of weak optimality, we arrive at

$$\mathbf{X}_*^\top \mathbf{A}^\dagger \mathbf{X}_* = \mathbf{U} \boldsymbol{\Sigma}_1^{1/2} \mathbf{Q}_1^\top (\mathbf{Q}_1 \boldsymbol{\Sigma}_1^{-1} \mathbf{Q}_1^\top + \mathbf{Q}_2 \boldsymbol{\Sigma}_2^{-1} \mathbf{Q}_2^\top) \mathbf{Q}_1 \boldsymbol{\Sigma}_1^{1/2} \mathbf{U}^\top \overset{(a)}{=} \mathbf{I}_r$$

where in (a) we use the facts $\mathbf{Q}_1^\top \mathbf{Q}_1 = \mathbf{I}_r$ and $\mathbf{Q}_1^\top \mathbf{Q}_2 = \mathbf{0}_{r \times (r_A - r)}$. This concludes the proof. $\quad\square$

### B.3.2 PROOF OF LEMMA 4

*Proof.* The proof is based on induction. First we have that $\mathbf{X}_0 = \mathbf{A}\boldsymbol{\Omega} = \mathbf{Q}\boldsymbol{\Sigma}\mathbf{Q}^\top\boldsymbol{\Omega}$. It is clear that $\boldsymbol{\Phi}_0 = \boldsymbol{\Sigma}\mathbf{Q}^\top\boldsymbol{\Omega}$. Now suppose that one can write $\mathbf{X}_t = \mathbf{Q}\boldsymbol{\Phi}_t$, following the update (2), it is not hard to see that

$$\begin{aligned}
\boldsymbol{\Phi}_{t+1} &= \boldsymbol{\Phi}_t - \eta\big(\boldsymbol{\Phi}_t\boldsymbol{\Phi}_t^\top - \boldsymbol{\Sigma}\big)\boldsymbol{\Phi}_t(\boldsymbol{\Phi}_t^\top\boldsymbol{\Phi}_t)^{-1} \\
&= (1-\eta)\boldsymbol{\Phi}_t + \eta\boldsymbol{\Sigma}\underbrace{\boldsymbol{\Phi}_t(\boldsymbol{\Phi}_t^\top\boldsymbol{\Phi}_t)^{-1}}_{:=\boldsymbol{\Theta}_t}.
\end{aligned} \tag{15}$$

The variable $\boldsymbol{\Theta}_t \in \mathbb{R}^{r_A \times r}$ can be roughly viewed as a pseudo-inverse of $\boldsymbol{\Phi}_t^\top$ because $\boldsymbol{\Phi}_t^\top\boldsymbol{\Theta}_t = \mathbf{I}_r$. We note that the invertibility of $(\boldsymbol{\Phi}_t^\top\boldsymbol{\Phi}_t)$ will become clear in Lemma 9. $\quad\square$

### B.3.3 PROOF OF THEOREM 2

*Proof.* Using $\boldsymbol{\Phi}_t^\top\boldsymbol{\Theta}_t = \mathbf{I}_r$, definition of $\mathbf{B}_t = \boldsymbol{\Phi}_t^\top\boldsymbol{\Sigma}^{-1}\boldsymbol{\Phi}_t$ (at the start of Apdx. B.3), and the update of $\boldsymbol{\Phi}_{t+1}$ in (15), it is not difficult to verify that

$$\mathbf{B}_{t+1} = (1-\eta)^2\mathbf{B}_t + 2\eta(1-\eta)\mathbf{I}_r + \eta^2\boldsymbol{\Theta}_t^\top\boldsymbol{\Sigma}\boldsymbol{\Theta}_t. \tag{16}$$

Subtracting $\mathbf{I}_r$ on both sides of (16), we can get

$$\mathbf{B}_{t+1} - \mathbf{I}_r = (1-\eta)^2(\mathbf{B}_t - \mathbf{I}_r) - \eta^2\mathbf{I}_r + \eta^2\boldsymbol{\Theta}_t^\top\boldsymbol{\Sigma}\boldsymbol{\Theta}_t.$$

This ensures that

$$\begin{aligned}
&\|\mathbf{B}_{t+1} - \mathbf{I}_r\|_{\mathsf{F}} \\
&\leq (1-\eta)^2\|\mathbf{B}_t - \mathbf{I}_r\|_{\mathsf{F}} + \eta^2\sqrt{r} + \eta^2\|\boldsymbol{\Theta}_t^\top\boldsymbol{\Sigma}\boldsymbol{\Theta}_t\|_{\mathsf{F}} \\
&\leq (1-\eta)^2\|\mathbf{B}_t - \mathbf{I}_r\|_{\mathsf{F}} + \eta^2\sqrt{r} + \eta^2\frac{r}{\sigma_r(\mathbf{B}_t)}
\end{aligned}$$

where the last inequality is because of Lemma 10. Suppose that $\eta \leq 2/3$, from Lemma 9, one can see that there exists a time $T_1$ such that $\sigma_r(\mathbf{B}_t) \geq 1/3, \forall t \geq T_1$. We assume $T_1 = 0$ following the same argument (i.e., initialized large with large $\xi$) as previous proofs. With these arguments, we obtain that

$$\begin{aligned}
&\|\mathbf{B}_{t+1} - \mathbf{I}_r\|_{\mathsf{F}} \\
&\leq (1-\eta)\|\mathbf{B}_t - \mathbf{I}_r\|_{\mathsf{F}} + \eta^2\sqrt{r} + 3r\eta^2 \\
&\leq \eta\sqrt{r} + 3\eta r + (1-\eta)^{t+1-T_1}\|\mathbf{B}_{T_1} - \mathbf{I}_r\|_{\mathsf{F}} \\
&\leq \eta\sqrt{r} + 3\eta r + (1-\eta)^{t+1-T_1}\|\mathbf{B}_{T_1} - \mathbf{I}_r\|_{\mathsf{F}}.
\end{aligned} \tag{17}$$

This implies a linear rate, i.e, $\|\mathbf{B}_{t+1} - \mathbf{I}_r\|_{\mathsf{F}} \leq \mathcal{O}(\eta r) + \epsilon$ if $\eta = \mathcal{O}(1)$ with sufficient iterations.

Inequality (17) also implies that choosing $\eta = \mathcal{O}(\epsilon/r)$, $\|\mathbf{B}_{t+1} - \mathbf{I}_r\|_{\mathsf{F}} \leq \epsilon$ at a rate of $\mathcal{O}(\frac{r}{\epsilon}\log\frac{1}{\epsilon})$. The proof is thus completed. $\quad\square$

### B.3.4 PROOF OF LEMMA 5

*Proof.* We start with notation. Let

$$\boldsymbol{\Sigma} = \begin{bmatrix} \boldsymbol{\Sigma}_1 & \mathbf{0} \\ \mathbf{0} & \boldsymbol{\Sigma}_2 \end{bmatrix}, \qquad \boldsymbol{\Phi}_t = \begin{bmatrix} \mathbf{M}_t \\ \mathbf{N}_t \end{bmatrix}, \tag{18}$$

where $\mathbf{\Sigma}_1 \in \mathbb{R}^{r \times r}$ is the learnable eigenvalues, while $\mathbf{\Sigma}_2 \in \mathbb{R}^{(r_A - r) \times (r_A - r)}$ are the unlearnable eigenvalues, and $\mathbf{M}_t \in \mathbb{R}^{r \times r}$ and $\mathbf{N}_t \in \mathbb{R}^{(r_A - r) \times r}$. Ideally at global convergence, we hope that $\mathbf{M}_t \to \mathbf{\Sigma}_1^{1/2}$ up to rotation; while $\mathbf{N}_t \to \mathbf{0}$.

We consider a scenario with $t \to \infty$, i.e., $\epsilon \to 0$ and $\mathbf{B}_t = \mathbf{I}_r$. Using (18) to rewrite $\mathbf{B}_t = \mathbf{I}_r$, we have that

$$\mathbf{M}_t^\top \mathbf{\Sigma}_1^{-1} \mathbf{M}_t + \mathbf{N}_t^\top \mathbf{\Sigma}_2^{-1} \mathbf{N}_t = \mathbf{I}_r. \tag{19}$$

The above equation implies that

$$\mathrm{Tr}(\mathbf{M}_t^\top \mathbf{\Sigma}_1^{-1} \mathbf{M}_t) = \mathrm{Tr}(\mathbf{M}_t^\top \mathbf{\Sigma}_1^{-1/2} \mathbf{\Sigma}_1^{-1/2} \mathbf{M}_t) \tag{20}$$

$$= \|\mathbf{\Sigma}_1^{-1/2} \mathbf{M}_t\|_\mathsf{F}^2 \overset{(a)}{\leq} r$$

where (a) is by (19) and Lemma 18.

Since we hope $\mathbf{\Sigma}_1^{-1/2} \mathbf{M}_t \to \mathbf{I}_r$, we have that

$$\begin{aligned}
& \|\mathbf{\Sigma}_1^{-1/2} \mathbf{M}_t - \mathbf{I}_r\|_\mathsf{F}^2 \\
&= \mathrm{Tr}\Big( (\mathbf{\Sigma}_1^{-1/2} \mathbf{M}_t - \mathbf{I}_r)^\top (\mathbf{\Sigma}_1^{-1/2} \mathbf{M}_t - \mathbf{I}_r) \Big) \\
&= \mathrm{Tr}\big( \mathbf{M}_t^\top \mathbf{\Sigma}_1^{-1/2} \mathbf{\Sigma}_1^{-1/2} \mathbf{M}_t \big) + \mathrm{Tr}(\mathbf{I}_r) - 2\mathrm{Tr}(\mathbf{M}_t^\top \mathbf{\Sigma}_1^{-1/2}) \\
&\overset{(a)}{\leq} \mathrm{Tr}\big( \mathbf{M}_t^\top \mathbf{\Sigma}_1^{-1/2} \mathbf{\Sigma}_1^{-1/2} \mathbf{M}_t \big) + \mathrm{Tr}(\mathbf{I}_r) + 2r^{3/2} \\
&\overset{(b)}{\leq} 2r + 2r^{3/2},
\end{aligned} \tag{21}$$

where (a) is because that i) for any $r \times r$ matrix $\mathbf{C}$, we have that $\mathrm{Tr}(\mathbf{C}) \geq r \min_i \mathbf{C}_{ii} \geq -r\|\mathbf{C}\|_\mathsf{F}$, ii) take $\mathbf{C} = \mathbf{M}_t^\top \mathbf{\Sigma}_1^{-1/2}$ and then apply (20); and (b) is by (20).

To bound $\mathbf{N}_t$, it can be seen that

$$\frac{1}{\sigma_{r+1}(\mathbf{A})} \mathrm{Tr}\big( \mathbf{N}_t^\top \mathbf{N}_t \big) \leq \mathrm{Tr}\big( \mathbf{N}_t^\top \mathbf{\Sigma}_2^{-1} \mathbf{N}_t \big) \overset{(c)}{\leq} r \tag{22}$$

where (c) is by applying Lemma 18 on (19). This suggests that $\|\mathbf{N}_t\|_\mathsf{F} \leq \sqrt{r\sigma_{r+1}(\mathbf{A})}$.

Lastly, note that $\mathbf{X}_*$ can be written as $\mathbf{X}_* = \mathbf{Q}[\mathbf{\Sigma}_1^{1/2}, \mathbf{0}]^\top$ and $\mathbf{X}_t = \mathbf{Q}\mathbf{\Phi}_t$. Using this fact and combining (21) and (22), we have that

$$\begin{aligned}
\|\mathbf{X}_t - \mathbf{X}_*\|_\mathsf{F}^2 &= \|\mathbf{M}_t - \mathbf{\Sigma}_1^{1/2}\|_\mathsf{F}^2 + \|\mathbf{N}_t\|_\mathsf{F}^2 \\
&= \|\mathbf{\Sigma}_1^{1/2}(\mathbf{\Sigma}_1^{-1/2} \mathbf{M}_t - \mathbf{I}_r)\|_\mathsf{F}^2 + \|\mathbf{N}_t\|_\mathsf{F}^2 \\
&\leq \sigma_1(\mathbf{\Sigma}_1^{1/2})^2 \|\mathbf{\Sigma}_1^{-1/2} \mathbf{M}_t - \mathbf{I}_r\|_\mathsf{F}^2 + \|\mathbf{N}_t\|_\mathsf{F}^2 \\
&= \mathcal{O}(r^{3/2}),
\end{aligned} \tag{23}$$

where we used $\sigma_1(\mathbf{\Sigma}) = 1$ and $\sigma_{r+1}(\mathbf{\Sigma}) \leq 1$. The proof is thus completed. $\qquad \square$

### B.3.5 USEFUL LEMMAS FOR SYMMETRIC AND UNDER-PARAMETRIZED PROBLEMS

It is clear that $\mathbf{B}_t$ is symmetric by definition, i.e., $\mathbf{B}_t = \mathbf{\Phi}_t^\top \mathbf{\Sigma}^{-1} \mathbf{\Phi}_t$. This enables us to give a lower bound on $\sigma_r(\mathbf{B}_t)$ using Lemma 15.

**Lemma 9.** $\sigma_r(\mathbf{B}_t)$ *is lower bounded by*

$$\sigma_r(\mathbf{B}_{t+1}) \geq (1 - \eta) - (1 - \eta)^{2t+3} + (1 - \eta)^{2t+2} \sigma_r(\mathbf{B}_0).$$

*Proof.* Given the definition of $\mathbf{B}_t$, it is not difficult to see that $\mathbf{B}_t$ is PSD for all $t$. We can then apply Lemma 15 on (16) to arrive at

$$
\begin{aligned}
&\sigma_r(\mathbf{B}_{t+1}) \\
&\geq 2\eta(1-\eta) + \sigma_r\big((1-\eta)^2\mathbf{B}_t + \eta^2\mathbf{\Theta}_t^\top\mathbf{\Sigma}\mathbf{\Theta}_t\big) \\
&\geq 2\eta(1-\eta) + (1-\eta)^2\sigma_r\big(\mathbf{B}_t\big) \\
&\overset{(a)}{\geq} (1-\eta)^{2t+2}\sigma_r(\mathbf{B}_0) + 2\eta(1-\eta)\frac{1-(1-\eta)^{2t+2}}{2\eta-\eta^2} \\
&\overset{(b)}{\geq} (1-\eta)^{2t+2}\sigma_r(\mathbf{B}_0) + (1-\eta) - (1-\eta)^{2t+3}
\end{aligned}
$$

where (a) uses Lemma 14 to unroll $\sigma_r(\mathbf{B}_t)$; and (b) is because $\frac{2\eta}{2\eta-\eta^2} \geq 1$. □

**Lemma 10.** *Let $\mathbf{\Theta}_t$ and $\mathbf{B}_t$ defined the same as those in Apdx. B.3. It is guaranteed to have that*

$$
\|\mathbf{\Theta}_t^\top\mathbf{\Sigma}\mathbf{\Theta}_t\|_\mathsf{F} \leq \frac{r}{\sigma_r(\mathbf{B}_t)}.
$$

*Proof.* Using the inequality $\|\mathbf{A}^\top\mathbf{A}\|_\mathsf{F} \leq \|\mathbf{A}\|_\mathsf{F}^2$, we have that

$$
\|\mathbf{\Theta}_t^\top\mathbf{\Sigma}\mathbf{\Theta}_t\|_\mathsf{F} = \|\mathbf{\Theta}_t^\top\mathbf{\Sigma}^{1/2}\mathbf{\Sigma}^{1/2}\mathbf{\Theta}_t\|_\mathsf{F} \leq \|\mathbf{\Sigma}^{1/2}\mathbf{\Theta}_t\|_\mathsf{F}^2. \tag{24}
$$

Now let $\mathbf{E}_t := \mathbf{\Sigma}^{1/2}\mathbf{\Theta}_t$ and $\mathbf{F}_t := \mathbf{\Sigma}^{-1/2}\mathbf{\Phi}_t$. Since we have that $\mathbf{F}_t^\top\mathbf{E}_t = \mathbf{I}_r$, we have that

$$
\|\mathbf{F}_t^\top\mathbf{E}_t\|_\mathsf{F} = \|\mathbf{I}_r\|_\mathsf{F} = \sqrt{r}.
$$

Since we also have that

$$
\sqrt{r} = \|\mathbf{F}_t^\top\mathbf{E}_t\|_\mathsf{F} \overset{(a)}{\geq} \sigma_r(\mathbf{F}_t)\|\mathbf{E}_t\|_\mathsf{F} \overset{(b)}{=} \sqrt{\sigma_r(\mathbf{B}_t)}\|\mathbf{E}_t\|_\mathsf{F}, \tag{25}
$$

where (a) holds because $\mathbf{E}_t$ and $\mathbf{F}_t$ share the same column space and row space and both of them have rank $r$, which implies that $\langle Null(\mathbf{F}), [\mathbf{E}_t]_i \rangle = \mathbf{0}, \forall i$ ($[\mathbf{E}_t]_i$ is the $i$th column of $\mathbf{E}_t$). Note that (a) does not hold true for general two matrices $\mathbf{E}_t$ and $\mathbf{F}_t$. (b) is because $\mathbf{F}_t^\top\mathbf{F}_t = \mathbf{B}_t$, which means that the singular values of $\mathbf{F}_t$ are just square root of eigenvalues of $\mathbf{B}_t$. This implies that $\|\mathbf{E}_t\|_\mathsf{F} \leq \sqrt{r}/\sqrt{\sigma_r(\mathbf{B}_t)}$. Combining this inequality with (24), we have that

$$
\|\mathbf{\Theta}_t^\top\mathbf{\Sigma}\mathbf{\Theta}_t\|_\mathsf{F} \leq \|\mathbf{\Theta}_t^\top\mathbf{\Sigma}^{1/2}\|_\mathsf{F}^2 = \|\mathbf{E}_t\|_\mathsf{F}^2 \leq \frac{r}{\sigma_r(\mathbf{B}_t)}.
$$

The proof is thus completed. □

### B.4 SYMMETRIC AND OVER-PARAMETRIZED SETTING

**Nyström initialization for over-parametrization.** While the initialization still follows (3), we need to adapt Lemma 1 to the over-parameterized setting, i.e., $r > r_A$.

**Lemma 11** (Initialization for over-parametrization). *There exists a universal constant $\tau > 0$ such that $\sigma_{r_A}(\mathbf{X}_0) \geq \xi\tau(\sqrt{r} - \sqrt{r_A-1})\sigma_{r_A}(\mathbf{A})$ is satisfied with high probability. In other words, $rank(\mathbf{X}_0) = r_A$ w.h.p.*

*Proof.* Similar to the proof of Lemma 1, let the compact eigenvalue decomposition of $\mathbf{A}$ be $\mathbf{A} = \mathbf{Q}\mathbf{\Sigma}\mathbf{Q}^\top$, where $\mathbf{Q} \in \mathbb{R}^{m \times r_A}$ and $\mathbf{\Sigma} \in \mathbb{R}^{r_A \times r_A}$. This implies that $\mathbf{X}_0 = (\mathbf{Q}\mathbf{\Sigma})(\mathbf{Q}^\top\mathbf{\Omega})$.

It is not hard to verify that the matrix $\mathbf{Q}^\top\mathbf{\Omega} \in \mathbb{R}^{r_A \times r}$ is also a Gaussian random matrix, where each entry follows $\mathcal{N}(0, \xi^2)$. Applying Lemma 19 on $(\mathbf{Q}^\top\mathbf{\Omega})^\top$, and using the fact $(\mathbf{Q}^\top\mathbf{\Omega})^\top$ and $(\mathbf{Q}^\top\mathbf{\Omega})$ share the same singular values, it can be seen that

$$
\mathbb{P}\Big(\frac{\sigma_{r_A}(\mathbf{Q}^\top\mathbf{\Omega})}{\xi} \leq \tau(\sqrt{r} - \sqrt{r_A-1})\Big) \leq (C_1\tau)^{r-r_A+1} + e^{-C_2 r} := \delta_2
$$

where $C_1$ and $C_2$ are universal constants independent of $r_A$ and $r$. This inequality shows that with probability at least $1 - \delta_2$, $\sigma_{r_A}(\mathbf{Q}^\top\mathbf{\Omega}) \geq \xi\tau(\sqrt{r} - \sqrt{r_A-1})$.

Note that inequality $\sigma_{\min}(\mathbf{CD}) \geq \sigma_{\min}(\mathbf{C})\sigma_{\min}(\mathbf{D})$ holds given full column rank of $\mathbf{C}$; see Lemma 17. Applying it to (9), we have that

$$\sigma_{r_A}(\mathbf{X}_0) \geq \sigma_{r_A}(\mathbf{Q}\mathbf{\Sigma})\sigma_{r_A}(\mathbf{Q}^\top\mathbf{\Omega}) = \sigma_{r_A}(\mathbf{A})\sigma_{r_A}(\mathbf{Q}^\top\mathbf{\Omega})$$

$$\overset{(a)}{\geq} \xi\tau(\sqrt{r} - \sqrt{r_A - 1})\sigma_{r_A}(\mathbf{A})$$

where (a) holds with probability at least $1 - \delta_2$. $\qquad\square$

Next, we provide additional results of Nyström initialization on over-paramtrized setting of problem (1), where we have $r_A < r$. For a desirable convergence rate, we need to slightly modify the ScaledGD update to

$$\mathbf{X}_{t+1} = \mathbf{X}_t - \eta(\mathbf{X}_t\mathbf{X}_t^\top - \mathbf{A})\mathbf{X}_t(\mathbf{X}_t^\top\mathbf{X}_t)^\dagger. \tag{26}$$

Compared with iteration (2) for exact-parametrization, the modification is on $(\mathbf{X}_t^\top\mathbf{X}_t)^\dagger$. This pseudo-inverse is necessary because $(\mathbf{X}_t^\top\mathbf{X}_t)$ is not necessarily invertible in the over-parametrized setting. We note that unlike previous work (Xu et al., 2023) which modifies the same term to $(\mathbf{X}_t^\top\mathbf{X}_t + \lambda\mathbf{I})^{-1}$, (26) does not need the damping parameter $\lambda\mathbf{I}$ in the preconditioner. We will observe shortly in Fig. 4 that the quadratic rate is not achieved with the damping factor.

Let the compact eigendecomposition of $\mathbf{A} = \mathbf{Q}\mathbf{\Sigma}\mathbf{Q}^\top$ for $\mathbf{Q} \in \mathbb{R}^{m \times r_A}$, and $\mathbf{\Sigma} \in \mathbb{R}^{r_A \times r_A}$. We can also establish that $\mathbf{X}_t$ affords a simpler representation.

**Lemma 12.** *Under the Nyström initialization* (3) *and iteration* (26)*, the variable* $\mathbf{X}_t$ *can be written as* $\mathbf{X}_t = \mathbf{Q}\mathbf{\Phi}_t$ *for some* $\mathbf{\Phi}_t \in \mathbb{R}^{r_A \times r}$*. Moreover, we have that*

$$\mathbf{\Phi}_{t+1} = (1 - \eta)\mathbf{\Phi}_t + \eta\mathbf{\Sigma}(\mathbf{\Phi}_t^\dagger)^\top. \tag{27}$$

*Proof.* We prove this by induction. Clearly, our initialization satisfies this because $\mathbf{X}_0 = \mathbf{A}\mathbf{\Omega} = \mathbf{Q}\mathbf{\Sigma}\mathbf{Q}^\top\mathbf{\Omega}$, i.e., $\mathbf{\Phi}_0 := \mathbf{\Sigma}\mathbf{Q}^\top\mathbf{\Omega}$. Now suppose that $\mathbf{X}_t = \mathbf{Q}\mathbf{\Phi}_t$ holds for $t$. We then show that $\mathbf{X}_{t+1} = \mathbf{Q}\mathbf{\Phi}_{t+1}$ to finish the induction. In particular, plugging $\mathbf{X}_t = \mathbf{Q}\mathbf{\Phi}_t$ into (26), we arrive at

$$\mathbf{X}_{t+1} = \mathbf{Q}\underbrace{\left[\mathbf{\Phi}_t - \eta(\mathbf{\Phi}_t\mathbf{\Phi}_t^\top - \mathbf{\Sigma})\mathbf{\Phi}_t(\mathbf{\Phi}_t^\top\mathbf{\Phi}_t)^\dagger\right]}_{:=\mathbf{\Phi}_{t+1}}.$$

Clearly, the term inside the brackets is $\mathbf{\Phi}_{t+1}$. The induction is thus finished.

Now we proof the second part of this lemma. Let the SVD of $\mathbf{\Phi}_t := \mathbf{U}_t\mathbf{\Sigma}_t\mathbf{V}_t^\top$, where $\mathbf{U}_t \in \mathbb{R}^{r_A \times r_A}$, $\mathbf{\Sigma}_t \in \mathbb{R}^{r_A \times r_A}$, and $\mathbf{V}_t \in \mathbb{R}^{r \times r_A}$. We note that $\mathbf{U}_t$ is unitary for this case. With the SVD, we have that $\mathbf{\Phi}_t\mathbf{\Phi}_t^\top = \mathbf{U}_t\mathbf{\Sigma}_t^2\mathbf{U}_t^\top$, and $(\mathbf{\Phi}_t^\top\mathbf{\Phi}_t)^\dagger = \mathbf{V}_t\mathbf{\Sigma}_t^{-2}\mathbf{V}_t^\top$. Plugging these into $\mathbf{\Phi}_{t+1}$ defined earlier, we arrive at

$$\mathbf{\Phi}_{t+1} = \mathbf{\Phi}_t - \eta(\mathbf{U}_t\mathbf{\Sigma}_t^2\mathbf{U}_t^\top - \mathbf{\Sigma})\mathbf{U}_t\mathbf{\Sigma}_t\mathbf{V}_t^\top\mathbf{V}_t\mathbf{\Sigma}_t^{-2}\mathbf{V}_t^\top$$

$$= \mathbf{\Phi}_t - \eta(\mathbf{U}_t\mathbf{\Sigma}_t^2\mathbf{U}_t^\top - \mathbf{\Sigma})\mathbf{U}_t\mathbf{\Sigma}_t^{-1}\mathbf{V}_t^\top$$

$$= \mathbf{\Phi}_t - \eta\mathbf{U}_t\mathbf{\Sigma}_t\mathbf{V}_t^\top + \eta\mathbf{\Sigma}\mathbf{U}_t\mathbf{\Sigma}_t^{-1}\mathbf{V}_t^\top$$

$$= (1 - \eta)\mathbf{\Phi}_t + \eta\mathbf{\Sigma}(\mathbf{\Phi}_t^\dagger)^\top.$$

This completes the proof. $\qquad\square$

Next, let $\mathbf{B}_t = \mathbf{\Phi}_t\mathbf{\Phi}_t^\top$. With (27) we have that

$$\mathbf{B}_{t+1} = (1 - \eta)^2\mathbf{\Phi}_t\mathbf{\Phi}_t^\top + \eta(1 - \eta)\mathbf{\Phi}_t\mathbf{\Phi}_t^\dagger\mathbf{\Sigma} + \eta(1 - \eta)\mathbf{\Sigma}(\mathbf{\Phi}_t^\dagger)^\top\mathbf{\Phi}_t^\top + \eta^2\mathbf{\Sigma}(\mathbf{\Phi}_t^\dagger)^\top\mathbf{\Phi}_t^\dagger\mathbf{\Sigma}$$

$$\overset{(a)}{=} (1 - \eta)^2\mathbf{B}_t + 2\eta(1 - \eta)\mathbf{\Sigma} + \eta^2\mathbf{\Sigma}(\mathbf{\Phi}_t^\dagger)^\top\mathbf{\Phi}_t^\dagger\mathbf{\Sigma} \tag{28}$$

$$\overset{(b)}{=} (1 - \eta)^2\mathbf{B}_t + 2\eta(1 - \eta)\mathbf{\Sigma} + \eta^2\mathbf{\Sigma}\mathbf{B}_t^{-1}\mathbf{\Sigma},$$

where in (a) we used the SVD of $\mathbf{\Phi}_t := \mathbf{U}_t\mathbf{\Sigma}_t\mathbf{V}_t^\top$, where $\mathbf{U}_t \in \mathbb{R}^{r_A \times r_A}$, $\mathbf{\Sigma}_t \in \mathbb{R}^{r_A \times r_A}$ and $\mathbf{V}_t \in \mathbb{R}^{r \times r_A}$, $\mathbf{\Phi}_t^\dagger = \mathbf{V}_t\mathbf{\Sigma}_t^{-1}\mathbf{U}_t^\top$, and $\mathbf{U}_t$ is unitary; and in (b) we assume that $\mathbf{B}_t$ is full rank. Note that this assumption can be easily verified given $\mathrm{rank}(\mathbf{B}_0) = r_A$; and the iteration on $\mathbf{B}_t$ (28) is

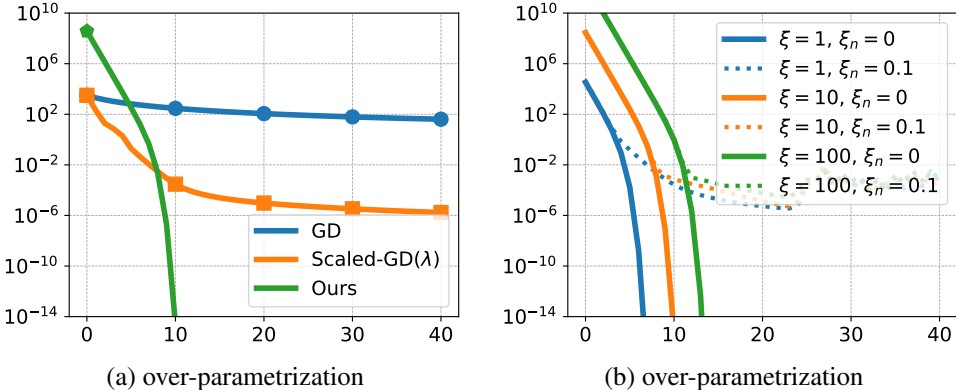

Figure 4: Convergence of ScaledGD under Nyström initialization (optimality error vs. iteration) on over-parametrized problems detailed in Apdx. E.1. (a) Comparison of GD, ScaledGD-($\lambda$) with small initialization, and ScaledGD with our initialization. (b) Solid lines show that our initialization is not sensitive to magnitude; and dotted lines illustrate that quadratic convergence cannot be obtained even with slightly perturbed initialization, i.e., $\mathbf{X}_0 = \mathbf{A\Omega} + \mathbf{N}$, where $[\mathbf{N}]_{ij} \sim \mathcal{N}(0, \xi_n^2)$.

exactly the same as in exact-parametrized cases (12). The latter allows us to bound $\sigma_{r_A}(\mathbf{B}_t)$ away from 0 in the same way as Lemma 2.

In other words, the over-parametrized case under our initialization reduces to the exact-parametrized case given the same iteration on $\mathbf{B}_t$ (28) (cf. (12)). This allows as to use the same argument of Theorem 1 to derive a quadratic rate for over-parametrized case.

**Theorem 5.** *With Nyström initialization* (3)*, the behavior of update* (26) *can be described as:*

*Phase 1 (linear convergence). Let* $\eta = \mathcal{O}(\frac{1}{\kappa^3 \|\mathbf{A}\|_\mathsf{F}})$*. After* $T_1 := \mathcal{O}(\kappa^3 \sqrt{r} \log \kappa)$ *iterations, ScaledGD ensures that* $\|\mathbf{X}_{T_1}\mathbf{X}_{T_1}^\top - \mathbf{A}\|_\mathsf{F} \leq \mathcal{O}(1/\kappa^2)$.

*Phase 2 (quadratic convergence). After Phase I, ScaledGD converges quadratically with* $\eta = 0.5$*. In particular,* $\|\mathbf{X}_T\mathbf{X}_T^\top - \mathbf{A}\|_\mathsf{F} \leq \epsilon$ *is ensured after* $T = \mathcal{O}\big(\log\log(\frac{1}{\kappa\epsilon})\big)$ *iterations.*

*Proof.* The proof is the same as Theorem 1 given the same iteration on $\mathbf{B}_t$ in (28). We omit it to avoid redundancy. □

**Numerical illustration.** A numerical illustration for ScaledGD under Nyström initialization in over-parametrized case can be found in Fig. 4. We adopt ScaledGD-($\lambda$) (Xu et al., 2023), the damping version of ScaledGD, as another baseline. It can be seen that only our approach achieves a quadratic rate; see Fig. 4(a). We also slightly perturb our initialization with small noise, and it can be seen that the quadratic convergence breaks down immediately. This demonstrate the critical role of initialization: i) it helps to get rid of damping using pseudo-inverse; and ii) it ensures a quadratic rate.

## C   MISSING PROOFS FOR ASYMMETRIC SETTINGS

### C.1   MISSING PROOFS FOR ASYMMETRIC AND EXACT-PARAMETRIZED SETTING

#### C.1.1   PROOF OF LEMMA 6

*Proof.* The proof is finished by induction. From our Nyström initialization, one has that $\mathbf{\Psi}_0 = \mathbf{0}$ and $\mathbf{\Phi}_0 = \mathbf{\Sigma V}^\top \mathbf{\Omega}$. Now assume that one can write $\mathbf{X}_t = \mathbf{U}\mathbf{\Phi}_t$ and $\mathbf{Y}_t = \mathbf{V}\mathbf{\Psi}_t$ for some iteration $t$. We will show that $\mathbf{X}_{t+1} = \mathbf{U}\mathbf{\Phi}_{t+1}$ and $\mathbf{Y}_{t+1} = \mathbf{V}\mathbf{\Psi}_{t+1}$ under iteration (6). Let us start with $\mathbf{X}_{t+1}$.

Note that if $t = 0$, $\mathbf{X}_1 = \mathbf{U}\boldsymbol{\Phi}_1$ is trivial. We only focus on $t \geq 1$, where we have

$$
\begin{aligned}
\mathbf{X}_{t+1} &= \mathbf{X}_t - \eta(\mathbf{X}_t \mathbf{Y}_t^\top - \mathbf{A})\mathbf{Y}_t(\mathbf{Y}_t^\top \mathbf{Y}_t)^{-1} \\
&= \mathbf{U}\boldsymbol{\Phi}_t - \eta(\mathbf{U}\boldsymbol{\Phi}_t \boldsymbol{\Psi}_t^\top \mathbf{V}^\top - \mathbf{U}\boldsymbol{\Sigma}\mathbf{V}^\top)\mathbf{V}\boldsymbol{\Psi}_t(\boldsymbol{\Psi}_t^\top \mathbf{V}^\top \mathbf{V}\boldsymbol{\Psi}_t)^{-1} \\
&= \mathbf{U}\boldsymbol{\Phi}_t - \eta\mathbf{U}(\boldsymbol{\Phi}_t \boldsymbol{\Psi}_t^\top - \boldsymbol{\Sigma})\boldsymbol{\Psi}_t(\boldsymbol{\Psi}_t^\top \boldsymbol{\Psi}_t)^{-1} \\
&= \mathbf{U}\underbrace{\left(\boldsymbol{\Phi}_t - \eta(\boldsymbol{\Phi}_t \boldsymbol{\Psi}_t^\top - \boldsymbol{\Sigma})\boldsymbol{\Psi}_t(\boldsymbol{\Psi}_t^\top \boldsymbol{\Psi}_t)^{-1}\right)}_{:=\boldsymbol{\Phi}_{t+1}}.
\end{aligned}
$$

Note that the invertible of $(\boldsymbol{\Psi}_t^\top \boldsymbol{\Psi}_t)$ will become clear in the proof of Corollary 1.

Using a similar argument, it is not hard to show that $\mathbf{Y}_t = \mathbf{V}\boldsymbol{\Psi}_t$ for all $t$. We do not repeat here. $\quad\square$

### C.1.2 PROOF OF THEOREM 3

*Proof.* Based on the initialization (5) and iteration (6), we can obtain that

$$
\boldsymbol{\Phi}_1 = \boldsymbol{\Phi}_0 \tag{29a}
$$

$$
\begin{aligned}
\boldsymbol{\Psi}_1 = \mathbf{V}^\top \mathbf{Y}_1 &= \mathbf{0} - \eta\mathbf{V}^\top(\mathbf{0} - \mathbf{A})^\top \mathbf{U}\boldsymbol{\Phi}_0(\boldsymbol{\Phi}_0^\top \mathbf{U}^\top \mathbf{U}\boldsymbol{\Phi}_0)^{-1} \\
&= \eta\mathbf{V}^\top \mathbf{V}\boldsymbol{\Sigma}\mathbf{U}^\top \mathbf{U}\boldsymbol{\Phi}_0(\boldsymbol{\Phi}_0^\top \mathbf{U}^\top \mathbf{U}\boldsymbol{\Phi}_0)^{-1} \\
&= \eta\boldsymbol{\Sigma}\boldsymbol{\Phi}_0(\boldsymbol{\Phi}_0^\top \boldsymbol{\Phi}_0)^{-1} \\
&= \eta\boldsymbol{\Sigma}\boldsymbol{\Phi}_0^{-\top}.
\end{aligned} \tag{29b}
$$

This ensures that

$$
\boldsymbol{\Phi}_1 \boldsymbol{\Psi}_1^\top = \eta\boldsymbol{\Sigma}.
$$

Choosing $\eta = 1$ completes the proof. $\quad\square$

### C.1.3 PROOF OF COROLLARY 1

*Proof.* The corollary is proved through an asymmetric-to-symmetric reduction.

**Step 1. Positive definiteness of $\boldsymbol{\Phi}_t \boldsymbol{\Psi}_t^\top$.** We will first show that $\boldsymbol{\Phi}_t \boldsymbol{\Psi}_t^\top$ is symmetric and positive definite (PD) for any $t \geq 1$. From the proof of Theorem 3, it can be seen that $\boldsymbol{\Phi}_1 \boldsymbol{\Psi}_1^\top = \eta\boldsymbol{\Sigma}$ is symmetric and PD. This means that the base case of induction holds. Now suppose that $\boldsymbol{\Phi}_t \boldsymbol{\Psi}_t^\top$ is symmetric and PD at iteration $t$. Based on Lemma 6, we can write the iteration as

$$
\boldsymbol{\Phi}_{t+1} = (1 - \eta)\boldsymbol{\Phi}_t + \eta\boldsymbol{\Sigma}\boldsymbol{\Psi}_t^{-\top} \tag{30a}
$$

$$
\boldsymbol{\Psi}_{t+1} = (1 - \eta)\boldsymbol{\Psi}_t + \eta\boldsymbol{\Sigma}\boldsymbol{\Phi}_t^{-\top}. \tag{30b}
$$

This gives that

$$
\boldsymbol{\Phi}_{t+1}\boldsymbol{\Psi}_{t+1}^\top = (1 - \eta)^2 \boldsymbol{\Phi}_t \boldsymbol{\Psi}_t^\top + 2\eta(1 - \eta)\boldsymbol{\Sigma} + \eta^2 \boldsymbol{\Sigma}(\boldsymbol{\Phi}_t \boldsymbol{\Psi}_t^\top)^{-1}\boldsymbol{\Sigma}. \tag{31}
$$

The symmetry of $\boldsymbol{\Phi}_{t+1}\boldsymbol{\Psi}_{t+1}^\top$ directly follows from (31). For the positive definiteness of $\boldsymbol{\Phi}_{t+1}\boldsymbol{\Psi}_{t+1}^\top$, we can apply Lemma 15 to get

$$
\lambda_{\min}(\boldsymbol{\Phi}_{t+1}\boldsymbol{\Psi}_{t+1}^\top) \geq (1 - \eta)^2 \lambda_{\min}(\boldsymbol{\Phi}_t \boldsymbol{\Psi}_t^\top) + 2\eta(1 - \eta)\lambda_{\min}(\boldsymbol{\Sigma}) + \eta^2 \lambda_{\min}(\boldsymbol{\Sigma}(\boldsymbol{\Phi}_t \boldsymbol{\Psi}_t^\top)^{-1}\boldsymbol{\Sigma}) > 0.
$$

This concludes the PD of $\boldsymbol{\Phi}_{t+1}\boldsymbol{\Psi}_{t+1}^\top$.

**Step 2.** Define $\mathbf{B}_t := \boldsymbol{\Phi}_t \boldsymbol{\Psi}_t^\top$, then (31) can be rewritten as

$$
\mathbf{B}_{t+1} = (1 - \eta)^2 \mathbf{B}_t + 2\eta(1 - \eta)\boldsymbol{\Sigma} + \eta^2 \boldsymbol{\Sigma}\mathbf{B}_t^{-1}\boldsymbol{\Sigma} \tag{32}
$$

which is exactly the same iteration as (12) for the symmetric exact-parametrized case. Based on the results from Step 1, that is, $\boldsymbol{\Phi}_{t+1}\boldsymbol{\Psi}_{t+1}^\top$ is symmetric and PD, we can apply the same analysis steps for symmetric exact-parametrized problems, i.e., Theorem 1 to get the bounds stated in this corollary. We do not repeat for conciseness. $\quad\square$

## C.2 MISSING PROOFS FOR ASYMMETRIC AND UNDER-PARAMETRIZED SETTING

### C.2.1 HOW GOOD IS WEAK OPTIMALITY?

**Lemma 13.** *Every global optimum for* (4) *is also weakly optimal.*

*Proof.* We start with rewriting the SVD of $\mathbf{A} = \mathbf{U}\boldsymbol{\Sigma}\mathbf{V}^\top$ as

$$\mathbf{A} = [\mathbf{U}_1, \mathbf{U}_2] \begin{bmatrix} \boldsymbol{\Sigma}_1 & \mathbf{0} \\ \mathbf{0} & \boldsymbol{\Sigma}_2 \end{bmatrix} \begin{bmatrix} \mathbf{V}_1^\top \\ \mathbf{V}_2^\top \end{bmatrix} = \mathbf{U}_1\boldsymbol{\Sigma}_1\mathbf{V}_1^\top + \mathbf{U}_2\boldsymbol{\Sigma}_2\mathbf{V}_2^\top \tag{33}$$

where $\mathbf{U}_1 \in \mathbb{R}^{m \times r}$ and $\mathbf{U}_2 \in \mathbb{R}^{m \times (r_A - r)}$ are the first $r$ and other columns of $\mathbf{U}$, respectively; $\boldsymbol{\Sigma}_1 \in \mathbb{R}^{r \times r}$ and $\boldsymbol{\Sigma}_2 \in \mathbb{R}^{(r - r_A) \times (r - r_A)}$ are diagonal matrices formed by the first $r$ and rest diagonal entries of $\boldsymbol{\Sigma}$; and $\mathbf{V}_1 \in \mathbb{R}^{n \times r}$ and $\mathbf{V}_2 \in \mathbb{R}^{n \times (r_A - r)}$ are the first $r$ and other columns of $\mathbf{V}$.

It is not hard to see that the optimal solutions of (1) are $\mathbf{X}_* = \mathbf{U}_1\boldsymbol{\Sigma}_1^{1/2}\mathbf{Q}$ and $\mathbf{Y}_* = \mathbf{V}_1\boldsymbol{\Sigma}_1^{1/2}\mathbf{Q}^{-\top}$, where $\mathbf{Q} \in \mathbb{R}^{r \times r}$ is any invertible matrix. Using these notation, we have that

$$\mathbf{Y}_*^\top \mathbf{A}^\dagger \mathbf{X}_* = \mathbf{Q}^{-1}\boldsymbol{\Sigma}_1^{1/2}\mathbf{V}_1^\top (\mathbf{V}_1\boldsymbol{\Sigma}_1^{-1}\mathbf{U}_1^\top + \mathbf{V}_2\boldsymbol{\Sigma}_2^{-1}\mathbf{U}_2^\top)\mathbf{U}_1\boldsymbol{\Sigma}_1^{1/2}\mathbf{Q}$$

$$\overset{(a)}{=} \mathbf{I}_r$$

where in (a) we use the facts $\mathbf{U}_1^\top \mathbf{U}_1 = \mathbf{I}_r$ and $\mathbf{U}_1^\top \mathbf{U}_2 = \mathbf{0}_{r \times (r_A - r)}$. This concludes the proof. $\qquad\square$

### C.2.2 PROOF OF THEOREM 4

*Proof.* The update in (6) ensures that

$$\boldsymbol{\Phi}_1 = \boldsymbol{\Phi}_0, \tag{34a}$$

$$\begin{aligned} \boldsymbol{\Psi}_1 = \mathbf{V}^\top \mathbf{Y}_1 &= \mathbf{0} - \eta\mathbf{V}^\top(\mathbf{0} - \mathbf{A})^\top \mathbf{U}\boldsymbol{\Phi}_0(\boldsymbol{\Phi}_0^\top \mathbf{U}^\top \mathbf{U}\boldsymbol{\Phi}_0)^{-1} \\ &= \eta\mathbf{V}^\top \mathbf{V}\boldsymbol{\Sigma}\mathbf{U}^\top \mathbf{U}\boldsymbol{\Phi}_0(\boldsymbol{\Phi}_0^\top \mathbf{U}^\top \mathbf{U}\boldsymbol{\Phi}_0)^{-1} \\ &= \eta\boldsymbol{\Sigma}\boldsymbol{\Phi}_0(\boldsymbol{\Phi}_0^\top \boldsymbol{\Phi}_0)^{-1} \\ &\overset{(a)}{:=} \eta\boldsymbol{\Sigma}\boldsymbol{\Theta}_0 \end{aligned} \tag{34b}$$

where in (a) we define $\boldsymbol{\Theta}_t := \boldsymbol{\Phi}_t(\boldsymbol{\Phi}_t^\top \boldsymbol{\Phi}_t)^{-1}$.

From the Definition 2, we can see that

$$\begin{aligned} \mathbf{Y}_1^\top \mathbf{A}^\dagger \mathbf{X}_1 &= \boldsymbol{\Psi}_1^\top \mathbf{V}^\top \mathbf{V}\boldsymbol{\Sigma}^{-1}\mathbf{U}^\top \mathbf{U}\boldsymbol{\Phi}_1 = \boldsymbol{\Psi}_1^\top \boldsymbol{\Sigma}^{-1}\boldsymbol{\Phi}_1 \\ &= \eta\boldsymbol{\Theta}_0^\top \boldsymbol{\Sigma}\boldsymbol{\Sigma}^{-1}\boldsymbol{\Phi}_0 = \eta\mathbf{I}_r. \end{aligned}$$

This means that when $\eta = 1$, generalized weak optimality can be achieved in one step for under-parametrized problems. $\qquad\square$

## C.3 ASYMMETRIC AND OVER-PARAMETRIZED SETTING

Next, we establish the one step convergence with Nyström initialization in the asymmetric over-parametrized setting, where $r_A < r$. We also need to slightly modify the ScaledGD update to

$$\mathbf{X}_1 = \mathbf{X}_0, \text{ and } \mathbf{X}_{t+1} = \mathbf{X}_t - \eta(\mathbf{X}_t\mathbf{Y}_t^\top - \mathbf{A})\mathbf{Y}_t(\mathbf{Y}_t^\top \mathbf{Y}_t)^\dagger, \forall t \geq 1 \tag{35a}$$

$$\mathbf{Y}_{t+1} = \mathbf{Y}_t - \eta(\mathbf{X}_t\mathbf{Y}_t^\top - \mathbf{A})^\top \mathbf{X}_t(\mathbf{X}_t^\top \mathbf{X}_t)^\dagger, \forall t \geq 0. \tag{35b}$$

Comparing with (6), the difference is that here we use pseudo-inverse to bypass the possible non-invertibility of $(\mathbf{X}_t^\top \mathbf{X}_t)$ and $(\mathbf{Y}_t^\top \mathbf{Y}_t)$ in the over-parametrized case. We also note that to the best of our knowledge, there is no previous result that establishes the convergence of ScaledGD (or its variants) for asymmetric over-parametrized problems.

**Theorem 6.** *Under Nyström initialization* (5)*, the modified ScaledGD iterations* (35) *converge globally in a single step, i.e.,* $\mathbf{X}_1 \mathbf{Y}_1^\top = \mathbf{A}$ *if the learning rate is chosen as* $\eta = 1$.

*Proof.* Let the compact eigendecomposition of $\mathbf{A} = \mathbf{U}\boldsymbol{\Sigma}\mathbf{V}^\top$ for $\mathbf{U} \in \mathbb{R}^{m \times r_A}$, $\boldsymbol{\Sigma} \in \mathbb{R}^{r_A \times r_A}$, and $\mathbf{V} \in \mathbb{R}^{n \times r_A}$.

The Nyström initialization ensures that $\mathbf{X}_0 = \mathbf{X}_1 = \mathbf{U}\boldsymbol{\Phi}_0$, where $\boldsymbol{\Phi}_0 \in \mathbb{R}^{r_A \times r}$ and clearly $\boldsymbol{\Phi}_0 = \boldsymbol{\Sigma}\mathbf{V}^\top \boldsymbol{\Omega}$. Using the expression of $\mathbf{X}_1$, iteration (35) gives that

$$\mathbf{Y}_1 = \eta \mathbf{V}\boldsymbol{\Sigma}\boldsymbol{\Phi}_0 (\boldsymbol{\Phi}_0^\top \boldsymbol{\Phi}_0)^\dagger.$$

Let the compact SVD of $\boldsymbol{\Phi}_0 := \mathbf{P}\mathbf{D}\mathbf{Q}^\top$, where $\mathbf{P} \in \mathbb{R}^{r_A \times r_A}$, $\mathbf{D} \in \mathbb{R}^{r_A \times r_A}$ and $\mathbf{Q} \in \mathbb{R}^{r \times r_A}$. Note that $\mathbf{P}$ is unitary. With the compact SVD of $\boldsymbol{\Phi}_0$, we have that $(\boldsymbol{\Phi}_0^\top \boldsymbol{\Phi}_0)^\dagger = \mathbf{Q}\mathbf{D}^{-2}\mathbf{Q}^\top$, which implies that

$$\mathbf{X}_1 \mathbf{Y}_1^\top = \eta \mathbf{U}\mathbf{P}\mathbf{D}\mathbf{Q}^\top \mathbf{Q}\mathbf{D}^{-2}\mathbf{Q}^\top \mathbf{Q}\mathbf{D}\mathbf{P}^\top \boldsymbol{\Sigma}\mathbf{V}^\top \overset{(a)}{=} \mathbf{U}\boldsymbol{\Sigma}\mathbf{V}^\top = \mathbf{A}$$

where (a) is because $\mathbf{P}$ is unitary and the choice of $\eta = 1$. $\qquad\square$

# D    OTHER USEFUL LEMMAS

**Lemma 14.** *Let* $A_{t+1} = (1-\theta)A_t + \beta$ *with some* $\alpha \in (0, 1)$ *and* $\beta \geq 0$*, then we have*

$$A_{t+1} = (1-\theta)^{t+1}A_0 + \beta \frac{1 - (1-\theta)^{t+1}}{\theta} \leq (1-\theta)^{t+1}A_0 + \frac{\beta}{\theta}.$$

*Proof.* The proof can be completed by simply unrolling $A_{t+1}$ and using the fact $1 + \alpha + \alpha^2 + \ldots + \alpha^t \leq \frac{1}{1-\alpha}$. $\qquad\square$

**Lemma 15.** *If* $\mathbf{A} \in \mathbb{R}^{n \times n}$ *and* $\mathbf{B} \in \mathbb{R}^{n \times n}$ *are positive semi-definite matrices, we have* $\lambda_{\min}(\mathbf{A} + \mathbf{B}) \geq \lambda_{\min}(\mathbf{A}) + \lambda_{\min}(\mathbf{B})$.

*Proof.* The smallest eigenvalue of $\mathbf{A} + \mathbf{B}$ can be expressed as

$$\lambda_{\min}(\mathbf{A} + \mathbf{B}) = \min_{\mathbf{x} \neq \mathbf{0}} \frac{\mathbf{x}^\top (\mathbf{A} + \mathbf{B})\mathbf{x}}{\mathbf{x}^\top \mathbf{x}} = \min_{\mathbf{x}_1 \neq \mathbf{0}, \mathbf{x}_1 = \mathbf{x}_2} \frac{\mathbf{x}_1^\top \mathbf{A}\mathbf{x}_1}{\mathbf{x}_1^\top \mathbf{x}_1} + \frac{\mathbf{x}_2^\top \mathbf{B}\mathbf{x}_2}{\mathbf{x}_2^\top \mathbf{x}_2}. \tag{36}$$

On the other hand, we also have that

$$\lambda_{\min}(\mathbf{A}) + \lambda_{\min}(\mathbf{B}) = \min_{\mathbf{x}_1 \neq \mathbf{0}, \mathbf{x}_2 \neq \mathbf{0}} \frac{\mathbf{x}_1^\top \mathbf{A}\mathbf{x}_1}{\mathbf{x}_1^\top \mathbf{x}_1} + \frac{\mathbf{x}_2^\top \mathbf{B}\mathbf{x}_2}{\mathbf{x}_2^\top \mathbf{x}_2}. \tag{37}$$

Because (36) is a constrained version of the minimization problem (37), they share the same objective, but (36) has shrinked feasible region. It is not difficult to see that $\lambda_{\min}(\mathbf{A} + \mathbf{B}) \geq \lambda_{\min}(\mathbf{A}) + \lambda_{\min}(\mathbf{B})$. The proof is thus completed. $\qquad\square$

**Lemma 16.** *Consider a sequence* $\{A_t\}_t$ *with* $A_t \geq 0, \forall t$. *If there exists* $\alpha$ *such that* $A_{t+1} \leq \alpha A_t^2$ *and* $A_0 \leq \frac{1}{2\alpha}$, *$A_t$ converges to* 0 *at a quadratic rate, i.e.,*

$$A_{t+1} \leq \frac{1}{\alpha} \frac{1}{2^{2^{t+1}}}.$$

*Proof.* Unrolling $A_{t+1}$, we get that

$$A_{t+1} \leq \alpha A_t^2 \leq \alpha^3 A_{t-1}^4 \leq \alpha^7 A_{t-2}^8 \leq \frac{1}{\alpha}(\alpha A_0)^{2^{t+1}} \leq \frac{1}{\alpha} \frac{1}{2^{2^{t+1}}}.$$

The proof is thus completed. $\qquad\square$

**Lemma 17.** *Let* $\mathbf{A} \in \mathbb{R}^{m \times n}$ *be a matrix with full column rank and* $\mathbf{B} \in \mathbb{R}^{n \times p}$ *be a non-zero matrix. Let* $\sigma_{\min}(\cdot)$ *be the smallest non-zero singular value. Then it holds that* $\sigma_{\min}(\mathbf{A}\mathbf{B}) \geq \sigma_{\min}(\mathbf{A})\sigma_{\min}(\mathbf{B})$.

*Proof.* Using the min-max principle for singular values,

$$
\begin{aligned}
\sigma_{\min}(\mathbf{AB}) &= \min_{\|\mathbf{x}\|=1, \mathbf{x}\in\text{ColSpan}(\mathbf{B})} \|\mathbf{ABx}\| \\
&= \min_{\|\mathbf{x}\|=1, \mathbf{x}\in\text{ColSpan}(\mathbf{B})} \left\|\mathbf{A}\frac{\mathbf{Bx}}{\|\mathbf{Bx}\|}\right\| \cdot \|\mathbf{Bx}\| \\
&\overset{(a)}{=} \min_{\|\mathbf{x}\|=1, \|\mathbf{y}\|=1, \mathbf{x}\in\text{ColSpan}(\mathbf{B}), \mathbf{y}\in\text{ColSpan}(\mathbf{B})} \|\mathbf{Ay}\| \cdot \|\mathbf{Bx}\| \\
&\geq \min_{\|\mathbf{y}\|=1, \mathbf{y}\in\text{ColSpan}(\mathbf{B})} \|\mathbf{Ay}\| \cdot \min_{\|\mathbf{x}\|=1, \mathbf{x}\in\text{ColSpan}(\mathbf{B})} \|\mathbf{Bx}\| \\
&\geq \min_{\|\mathbf{y}\|=1} \|\mathbf{Ay}\| \cdot \min_{\|\mathbf{x}\|=1, \mathbf{x}\in\text{ColSpan}(\mathbf{B})} \|\mathbf{Bx}\| \\
&= \sigma_{\min}(\mathbf{A})\sigma_{\min}(\mathbf{B})
\end{aligned}
$$

where (a) is by changing of variables, i.e., $\mathbf{y} = \mathbf{Bx}/\|\mathbf{Bx}\|$. $\qquad\square$

**Lemma 18.** *For PSD matrices $\mathbf{A}$ and $\mathbf{B}$, if $\mathbf{A} + \mathbf{B} = \mathbf{I}_r$, then we have $Tr(\mathbf{A}) \leq r$ and $Tr(\mathbf{B}) \leq r$.*

*Proof.* The proof is straightforward and is omitted here. $\qquad\square$

**Lemma 19** (Rudelson & Vershynin (2009))**.** *Let $\mathbf{W}$ be an $d \times r$ matrix with $d \geq r$. The entries of $\mathbf{W}$ are drawn independently from $\mathcal{N}(0,1)$. Then for every $\tau > 0$, we have that*

$$
\mathbb{P}\big(\sigma_r(\mathbf{W}) \leq \tau(\sqrt{d} - \sqrt{r-1})\big) \leq (C_1\tau)^{d-r+1} + e^{-C_2 d}.
$$

*where $C_1$ and $C_2$ are universal constants independent of $d$ and $r$.*

# E MISSING EXPERIMENTAL DETAILS

## E.1 DETAILS FOR PROBLEMS WITH SYNTHETIC DATA

This subsection contains the detailed setup for the problems with synthetic data in Figs. 1 and 4. Recall that here we focus on symmetric problems under exact-, under-, and over-parametrization.

For the exact-parametrized problem in Fig. 1 (a) and (b), we choose the PSD matrix $\mathbf{A} \in \mathbb{R}^{m\times m}$ in the following manner. We set $m = 1000$ and $r = r_A = 20$. The non-zero singular values are set as $\{1.0, 0.99, 0.98, \dots, 0.82, 0.01\}$, where we intentionally set $\sigma_{r_A} = 0.01$ to enlarge the condition number. We choose the step size of GD as $0.01$ to avoid divergence. The learning rate for ScaledGD is $0.5$.

For the under-parametrized problem in Fig. 1 (c), we choose PSD matrix $\mathbf{A} \in \mathbb{R}^{m\times m}$ in the following manner. We set $m = 1000$ and $r_A = 40$. The singular values of $\mathbf{A}$ are $\{1.0, 0.99, 0.98, \dots, 0.65, 0.64, 0.05, 0.025, 0.01\}$. We choose $r = 20$ to ensure the under-parametrized nature of this problem.

For the over-parametrized case in Fig. 4 (a) and (b), we choose PSD matrix $\mathbf{A} \in \mathbb{R}^{m\times m}$ in the following manner. We set $m = 1000$ and $r_A = 20$. The non-zero singular values are chosen as $\{1.0, 0.99, 0.98, \dots, 0.82, 0.01\}$, where we intentionally set $\sigma_{r_A} = 0.01$ to enlarge the condition number. We set $\mathbf{X}$ to be over-parametrized by letting $r = 60$. We choose the step size of GD as $0.01$. The learning rate of ScaledGD-$\lambda$ is set as $0.5$, and its damping parameter $\lambda$ is chosen as $0.01$. The learning rate for ScaledGD with Nyström initialization is $0.5$.

## E.2 DATASETS

The evaluation of NoRA and NoRA+ is carried out on commonly adopted datasets in the literature.

**GLUE benchmark.** GLUE is designed to provide general-purpose evaluation of language understanding (Wang et al., 2019b). Those adopted in our work include SST-2 (sentiment analysis, (Socher et al., 2013)), RTE[4] (inference). These datasets are released under different permissive licenses.

---

[4]https://paperswithcode.com/dataset/rte

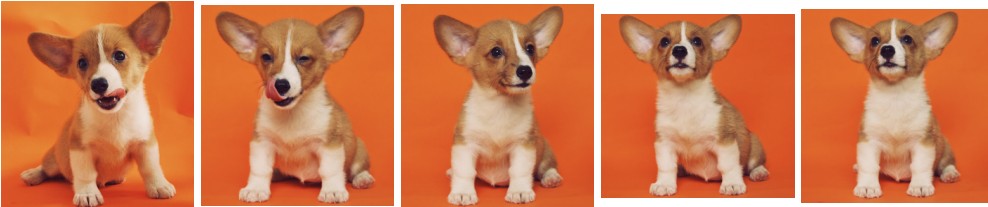

Figure 5: The dog dataset.

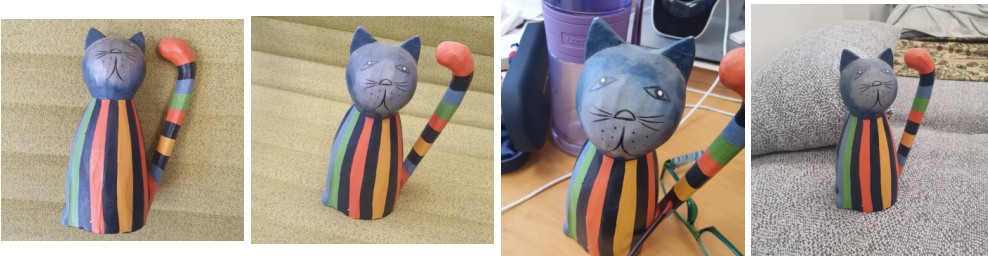

Figure 6: The cat-toy dataset.

**SuperGLUE benchmark.** SuperGLUE (Wang et al., 2019a) is another commonly adopted benchmark for language understanding, and it is more challenging compared with GLUE. The considered datasets include CB (inference, (De Marneffe et al., 2019)), ReCoRD (question answering, (Zhang et al., 2018)), WSC (coreference resolution, (Levesque et al., 2012)), BoolQ (question answering, (Clark et al., 2019)), and MiltiRC (question answering, (Khashabi et al., 2018)). These datasets are released under different permissive licenses.

**Commonsense reasoning.** These datasets are a collection tasks that require commonsense reasoning to answer. The considered datasets include WinoGrande (Sakaguchi et al., 2021), PIQA (Bisk et al., 2020), SOCIAL-I-QA (SIQA) (Sap et al., 2019), HellaSwag (Zellers et al., 2019), ARC-easy, ARC-challenge (Chollet, 2019) and OpenbookQA (Mihaylov et al., 2018). These datasets are released under different permissive licenses.

**Math.** For mathematical problems, we consider GSM8K (Cobbe et al., 2021) dataset that consists of high quality linguistically diverse school math problems created by human problem writers. This dataset is under MIT license. We also adopt MetaMathQA dataset (Yu et al., 2024), which is constructed through bootstrapping mathematical questions by rewriting the question from multiple perspectives. This dataset is under MIT license.

**Additional datasets.** We also use SQuAD (question answering, (Rajpurkar et al., 2016)) in our experiments, which is released under license CC BY-SA 4.0.

**Datasets for DreamBooth.** The datasets (dog and cat-toy) used for Sec. 5.2 are obtained directly from Huggingface. The dog dataset[5] contains 5 dog images; see Fig. 5. The cat-toy[6] dataset has 4 images; see Fig. 6. Both datasets are representative examples for the purpose of DreamBooth – finetuning with only few images for personalized generalization.

### E.3 DETAILS FOR FIG. 2

The experiment setting and training protocols are the same as few-shot learning with OPT-1.3B in the following subsection. Here, we are interested in the change of singular values after LoRA finetuning. For each LoRA layer, we compare the singular values of $\mathbf{W}_0$ and $\mathbf{W}_0 + \mathbf{X}_T \mathbf{Y}_T^\top$, where $\mathbf{X}_T, \mathbf{Y}_T$ are LoRA weights after training, and find out the indices of $r$ singular values that have the largest change after finetuning. We then count the indices across all LoRA layers. Fig. 2 plots indices vs. counts.

---

[5] https://huggingface.co/datasets/diffusers/dog-example
[6] https://huggingface.co/datasets/diffusers/cat-toy-example

### E.4 FEW-SHOT LEARNING WITH OPT-1.3B

For this experiment, we first search for the best batchsizes for LoRA, and the same batchsize is applied for other tested algorithms as well. Then we search additionally for the best learning rate for each algorithm. This ensures that different algorithms see the same amount of data, while still having their best performed learning rate. The hyperparameters adopted are searched over values in Tab. 5. Adam is adopted for optimization.

Table 5: Hyperparameters used for few-shot learning with OPT-1.3B.

| Hyperparameters | Values |
|---|---|
| LoRA $r$ | 8 |
| LoRA $\alpha$ | 16 |
| LoRA module | q_proj, v_proj |
| # epochs | 5 |
| batchsize | 2, 4, 8 |
| learning rate | $1\times10^{-5}$, $5\times10^{-5}$, $1\times10^{-4}$ |
| NoRA $\xi$ | 0.05, 0.1, 0.2 |

### E.5 DREAMBOOTH WITH STABLE-DIFFUSION

Stable Diffusion V1.4 (Rombach et al., 2022) is adopted as base model, where LoRA is applied to the UNet. The text-encoder is not finetuned. We adopt the default parameter-choice from Huggingface, which is summarized in Tab. 6. We adopt AdamW as the optimizer with a weight decay of 0.01.

Table 6: Hyperparameters used for DreamBooth with stable-diffusion.

| Hyperparameters | Values |
|---|---|
| LoRA $r$ | 4 |
| LoRA $\alpha$ | 4 |
| LoRA module | to_q, to_k, to_v, to_out |
| # iterations | 500 |
| batchsize | 1 |
| learning rate | $1\times10^{-4}$ |
| NoRA $\xi$ | 0.1 |

We provide additional results to further support the efficiency of NoRA by finetuning the stable-diffusion-v1.4 model using the same protocol as in Sec. 5.2. Here we adopt a dataset with 4 toy-cat images; see Fig. 6. After finetuning 500 steps using prompt "a photo of toy cat", our goal is to generate images "a toy cat wearing glasses." The generated images are shown in Fig. 7. In general, all tested algorithms do not distinguish the hands and the tail of toy cat well. However, both LoRA and LoRA-P generate images with less accurate facial details. For example, the glasses are not wearing well, or the eyes are not clear. However, the details of faces generated by NoRA and NoRA+ are quite clear.

### E.6 COMMONSENSE REASONING WITH LLAMA AND LLAMA2

The base models considered are LLaMA-7B and LLaMA2-7B. The experimental setup and choices of hyperparameters follow (Liu et al., 2024). The hyperparameters are summarized in Tab. 7.

### E.7 MATH REASONING WITH GEMMA-7B

Our last evaluation tackles mathematical reasoning. Gemma-7B (Gemma-team et al., 2024) is finetuned for 2 epochs on MetaMathQA-100K dataset (Yu et al., 2024). LoRA rank is set as 32, leading to 100M trainable parameters. The performance is assessed on GSM8K (Cobbe et al., 2021), and hyperparameters are summarized in Tab. 8.

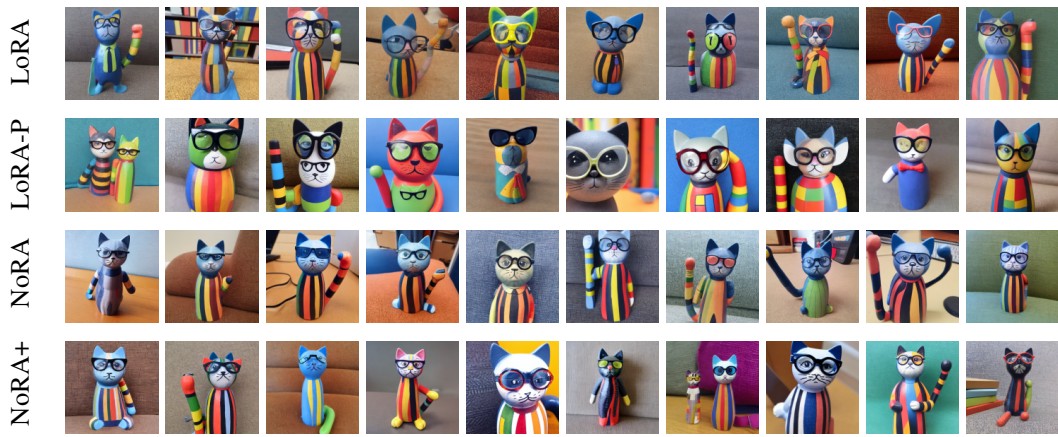

Figure 7: Generated images from NoRA and NoRA+ with stable-diffusion.

Table 7: Hyperparameters used for commonsense reasoning with LLaMA-7B and LLaMA2-7B.

| Hyper-parameters | Values |
|---|---|
| LoRA $r$ (rank) | 32 |
| LoRA $\alpha$ | 64 |
| LoRA module | q_proj, k_proj, v_proj, up_proj, down_proj |
| epoch | 3 |
| learning rate | $3 \times 10^{-4}$ |
| batchsize | 16 |
| cutoff length | 256 |
| NoRA $\xi$ | 0.02, 0.05, 0.1 |

The performance of various approaches is summarized in Tab. 9. We also include PiSSA (Meng et al., 2024) into the comparison. Note that PiSSA uses LoRA rank as 64 but is only finetuned for a single epoch. Despite this difference, the computational cost on backward passes is the same for PiSSA and NoRA. The results clearly show that NoRA (NoRA+) outperforms LoRA (LoRA-P), highlighting the effectiveness of our Nyström initialization.

Table 8: Hyperparameters used for math reasoning with Gemma-7B.

| Hyper-parameters | Values |
|---|---|
| LoRA $r$ (rank) | 32 |
| LoRA $\alpha$ | 64 |
| LoRA module | q_proj, k_proj, v_proj, o_proj, up_proj, down_proj, gate_proj |
| epoch | 2 |
| learning rate | $3 \times 10^{-4}, 4 \times 10^{-4}, 5 \times 10^{-4}$ |
| batchsize | 128 |
| NoRA $\xi$ | 0.02, 0.05, 0.1 |

Table 9: Performances of different algorithms for math reasoning tasks. The results marked with ‡ are taken from (Meng et al., 2024).

| GSM8K | LoRA | PiSSA[‡] | NoRA | LoRA-P | NoRA+ |
|---|---|---|---|---|---|
| Gemma-7B | 76.72 | 77.94 | 78.62 | 77.03 | 78.47 |

