# On the Crucial Role of Initialization for Matrix Factorization

## Abstract

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

$$\mathbf{\Psi}_{t+1} = (1-\eta)\mathbf{\Psi}_t + \eta\mathbf{\Sigma}\mathbf{\Phi}_t^{-\top}. \tag{30b}$$

This gives that

$$\mathbf{\Phi}_{t+1}\mathbf{\Psi}_{t+1}^\top = (1-\eta)^2\mathbf{\Phi}_t\mathbf{\Psi}_t^\top + 2\eta(1-\eta)\mathbf{\Sigma} + \eta^2\mathbf{\Sigma}(\mathbf{\Phi}_t\mathbf{\Psi}_t^\top)^{-1}\mathbf{\Sigma}. \tag{31}$$

The symmetry of $\mathbf{\Phi}_{t+1}\mathbf{\Psi}_{t+1}^\top$ directly follows from (31). For the positive definiteness of $\mathbf{\Phi}_{t+1}\mathbf{\Psi}_{t+1}^\top$, we can apply Lemma 15 to get

$$\lambda_{\min}(\mathbf{\Phi}_{t+1}\mathbf{\Psi}_{t+1}^\top) \geq (1-\eta)^2\lambda_{\min}(\mathbf{\Phi}_t\mathbf{\Psi}_t^\top) + 2\eta(1-\eta)\lambda_{\min}(\mathbf{\Sigma}) + \eta^2\lambda_{\min}(\mathbf{\Sigma}(\mathbf{\Phi}_t\mathbf{\Psi}_t^\top)^{-1}\mathbf{\Sigma}) > 0.$$

This concludes the PD of $\mathbf{\Phi}_{t+1}\mathbf{\Psi}_{t+1}^\top$.

**Step 2.** Define $\mathbf{B}_t := \mathbf{\Phi}_t\mathbf{\Psi}_t^\top$, then (31) can be rewritten as

$$\mathbf{B}_{t+1} = (1-\eta)^2\mathbf{B}_t + 2\eta(1-\eta)\mathbf{\Sigma} + \eta^2\mathbf{\Sigma}\mathbf{B}_t^{-1}\mathbf{\Sigma} \tag{32}$$

which is exactly the same iteration as (12) for the symmetric exact-parametrized case. Based on the results from Step 1, that is, $\mathbf{\Phi}_{t+1}\mathbf{\Psi}_{t+1}^\top$ is symmetric and PD, we can apply the same analysis steps for symmetric exact-parametrized problems, i.e., Theorem 1 to get the bounds stated in this corollary. We do not repeat for conciseness. □

### C.2 Missing proofs for asymmetric and under-parametrized setting

#### C.2.1 How good is weak optimality?

**Lemma 13.** *Every global optimum for* (4) *is also weakly optimal.*

*Proof.* We start with rewriting the SVD of $\mathbf{A} = \mathbf{U}\mathbf{\Sigma}\mathbf{V}^\top$ as

$$\mathbf{A} = [\mathbf{U}_1, \mathbf{U}_2]\begin{bmatrix}\mathbf{\Sigma}_1 & \mathbf{0} \\ \mathbf{0} & \mathbf{\Sigma}_2\end{bmatrix}\begin{bmatrix}\mathbf{V}_1^\top \\ \mathbf{V}_2^\top\end{bmatrix} = \mathbf{U}_1\mathbf{\Sigma}_1\mathbf{V}_1^\top + \mathbf{U}_2\mathbf{\Sigma}_2\mathbf{V}_2^\top \tag{33}$$

where $\mathbf{U}_1 \in \mathbb{R}^{m\times r}$ and $\mathbf{U}_2 \in \mathbb{R}^{m\times(r_A-r)}$ are the first $r$ and other columns of $\mathbf{U}$, respectively; $\mathbf{\Sigma}_1 \in \mathbb{R}^{r\times r}$ and $\mathbf{\Sigma}_2 \in \mathbb{R}^{(r-r_A)\times(r-r_A)}$ are diagonal matrices formed by the first $r$ and rest diagonal entries of $\mathbf{\Sigma}$; and $\mathbf{V}_1 \in \mathbb{R}^{n\times r}$ and $\mathbf{V}_2 \in \mathbb{R}^{n\times(r_A-r)}$ are the first $r$ and other columns of $\mathbf{V}$.

It is not hard to see that the optimal solutions of (1) are $\mathbf{X}_* = \mathbf{U}_1\mathbf{\Sigma}_1^{1/2}\mathbf{Q}$ and $\mathbf{Y}_* = \mathbf{V}_1\mathbf{\Sigma}_1^{1/2}\mathbf{Q}^{-\top}$, where $\mathbf{Q} \in \mathbb{R}^{r\times r}$ is any invertible matrix. Using these notation, we have that

$$\mathbf{Y}_*^\top\mathbf{A}^\dagger\mathbf{X}_* = \mathbf{Q}^{-1}\mathbf{\Sigma}_1^{1/2}\mathbf{V}_1^\top(\mathbf{V}_1\mathbf{\Sigma}_1^{-1}\mathbf{U}_1^\top + \mathbf{V}_2\mathbf{\Sigma}_2^{-1}\mathbf{U}_2^\top)\mathbf{U}_1\mathbf{\Sigma}_1^{1/2}\mathbf{Q}$$

$$\overset{(a)}{=} \mathbf{I}_r$$

where in (a) we use the facts $\mathbf{U}_1^\top\mathbf{U}_1 = \mathbf{I}_r$ and $\mathbf{U}_1^\top\mathbf{U}_2 = \mathbf{0}_{r\times(r_A-r)}$. This concludes the proof.

□

#### C.2.2 Proof of Theorem 4

*Proof.* The update in (6) ensures that

$$\mathbf{\Phi}_1 = \mathbf{\Phi}_0, \tag{34a}$$

$$\begin{aligned}\mathbf{\Psi}_1 = \mathbf{V}^\top\mathbf{Y}_1 &= \mathbf{0} - \eta\mathbf{V}^\top(\mathbf{0} - \mathbf{A})^\top\mathbf{U}\mathbf{\Phi}_0(\mathbf{\Phi}_0^\top\mathbf{U}^\top\mathbf{U}\mathbf{\Phi}_0)^{-1} \\ &= \eta\mathbf{V}^\top\mathbf{V}\mathbf{\Sigma}\mathbf{U}^\top\mathbf{U}\mathbf{\Phi}_0(\mathbf{\Phi}_0^\top\mathbf{U}^\top\mathbf{U}\mathbf{\Phi}_0)^{-1} \\ &= \eta\mathbf{\Sigma}\mathbf{\Phi}_0(\mathbf{\Phi}_0^\top\mathbf{\Phi}_0)^{-1} \\ &\overset{(a)}{:=} \eta\mathbf{\Sigma}\mathbf{\

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

 \mathrm{ColSpan}(\mathbf{B}), \mathbf{y} \in \mathrm{ColSpan}(\mathbf{B})} \|\mathbf{A}\mathbf{y}\| \cdot \|\mathbf{B}\mathbf{x}\|$$

$$\geq \min_{\|\mathbf{y}\|=1, \mathbf{y} \in \mathrm{ColSpan}(\mathbf{B})} \|\mathbf{A}\mathbf{y}\| \cdot \min_{\|\mathbf{x}\|=1, \mathbf{x} \in \mathrm{ColSpan}(\mathbf{B})} \|\mathbf{B}\mathbf{x}\|$$

$$\geq \min_{\|\mathbf{y}\|=1} \|\mathbf{A}\mathbf{y}\| \cdot \min_{\|\mathbf{x}\|=1, \mathbf{x} \in \mathrm{ColSpan}(\mathbf{B})} \|\mathbf{B}\mathbf{x}\|$$

$$= \sigma_{\min}(\mathbf{A})\sigma_{\min}(\mathbf{B})$$

where (a) is by changing of variables, i.e., $\mathbf{y} = \mathbf{B}\mathbf{x}/\|\mathbf{B}\mathbf{x}\|$. $\qquad\square$

**Lemma 18.** *For PSD matrices $\mathbf{A}$ and $\mathbf{B}$, if $\mathbf{A} + \mathbf{B} = \mathbf{I}_r$, then we have $Tr(\mathbf{A}) \leq r$ and $Tr(\mathbf{B}) \leq r$.*

*Proof.* The proof is straightforward and is omitted here. $\qquad\square$

**Lemma 19** (Rudelson & Vershynin (2009))**.** *Let $\mathbf{W}$ be an $d \times r$ matrix with $d \geq r$. The entries of $\mathbf{W}$ are drawn independently from $\mathcal{N}(0, 1)$. Then for every $\tau > 0$, we have that*

$$\mathbb{P}\left(\sigma_r(\mathbf{W}) \leq \tau(\sqrt{d} - \sqrt{r-1})\right) \leq (C_1 \tau)^{d-r+1} + e^{-C_2 d}.$$

*where $C_1$ and $C_2$ are universal constants independent of $d$ and $r$.*

# E   MISSING EXPERIMENTAL DETAILS

## E.1   DETAILS FOR PROBLEMS WITH SYNTHETIC DATA

This subsection contains the detailed setup for the problems with synthetic data in Figs. 1 and 4. Recall that here we focus on symmetric problems under exact-, under-, and over-parametrization.

For the exact-parametrized problem in Fig. 1 (a) and (b), we choose the PSD matrix $\mathbf{A} \in \mathbb{R}^{m \times m}$ in the following manner. We set $m = 1000$ and $r = r_A = 20$. The non-zero singular values are set as $\{1.0, 0.99, 0.98, \ldots, 0.82, 0.01\}$, where we intentionally set $\sigma_{r_A} = 0.01$ to enlarge the condition

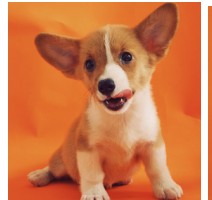 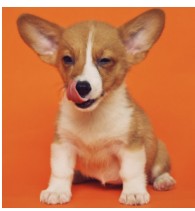 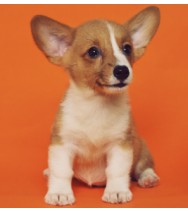 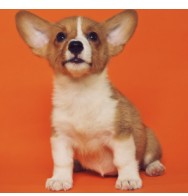 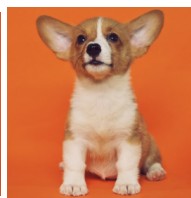

Figure 5: The dog dataset.

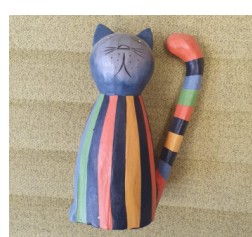 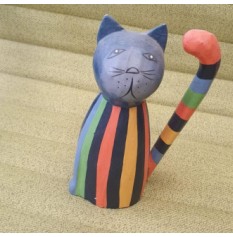 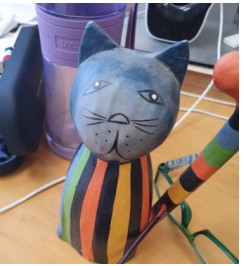 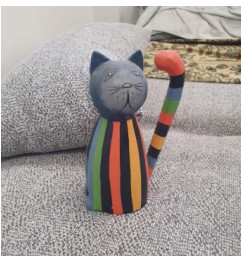

Figure 6: The cat-toy dataset.