# OpenReview forum: "On the Crucial Role of Initialization for Matrix Factorization"
_ICLR.cc/2025/Conference — ICLR 2025 Poster_

### Official Review · Reviewer_kgvD · 2024-10-22

**Soundness:** 3
**Presentation:** 3
**Contribution:** 3
**Rating:** 8
**Confidence:** 3

**Summary:**

This paper considers the matrix factorization problem, where one is given a matrix $A$ and the goal is to compute a rank-$r$ factorization that approximates $A$. It studies the setting when $A$ is PSD and the factorization is also PSD, and when $A$ is not symmetric. Three settings are studied: exact, over and under parametrization, meaning the rank of $A$ is equal, smaller or larger than $r$. Their algorithm is preconditioned gradient, and the key of this paper it show that a good initialization significantly improves convergence rate. The initialization is to set $X_0=A \Omega$ for a zero-mean Gaussian matrix $\Omega$. Due to rotational invariance of Gaussian, this initialization has many desired property, such as it aligns with the desired top-$r$ eigenspace. Using this technique, they obtain improved rates for both symmetric matrix factorization and asymmetric variants. They also show a major application for this theory, that is, the initialization for LoRA training. Standard LoRA setting initializes $X_0$ to random Gaussian matrix, and the theory developed in this paper suggests a better alternative is to use $X_0=W_0 \Omega$ where $W_0$ is the pre-trained weights. Experiments are performed on various few-shot learning tasks on benchmarks to show advantages over LoRA and its variants.

**Strengths:**

This paper presents a simple initialization framework for matrix factorization that significantly improves convergence rates, which is to align the principle directions of $X_0$ with $A$ through a rank-$r$ Gaussian. Algorithmically, this does not incur (asymptotically) extra cost as multiplying $A$ with a small matrix is cheap. It also showcases an interesting phenomenon: in the exact parametrization setting, the proposed algorithm has a linear convergence for iterations proportion to $\kappa^3$, then rapidly switches to quadratic convergence, resembles that of second-order method without explicitly resorting to second-order information. The application to LoRA is nice and natural.

**Weaknesses:**

I think the paper is overall well-written with no obvious weaknesses. The empirical improvements over LoRA seem rather marginal but I'm not an expert on experiments.

**Questions:**

What is the role of Gaussian playing in your analysis? To put it in other words, do you think it's possible to replace Gaussian with something that can be applied faster to $A$, or Gaussian is essential for the analysis?

---

> ### Author Response · Authors · 2024-11-18
>
> We thank the reviewer for recognizing the strengths of our paper. Below, we provide further discussions to address your questions.
>
> **W1.** The empirical improvements over LoRA seem rather marginal but I'm not an expert on experiments.
>
> **A1.** We hope to clarify our numerical results.
>
> - Sec. 5.1 shows that the proposed approach outperforms existing initialization approaches for LoRA. We note that the compared approaches, PiSSA and OLoRA have been integrated into HuggingFace's PEFT package [1], and are actively employed in the community.
>
> - Our results in Sec. 5.2 show that NoRA reduces the training loss of LoRA by 10\%. (This is an image generation task, hence we use loss to measure efficiency.)
>
> - For results in Sec. 5.3, note that LLaMA2 improves over LLaMA by 1.4 (in absolute accuracy). This is through significantly improved data quality and thousands of extra GPU hours. Our NoRA improves accuracy by 0.5 on LLaMA and 1.2 on LLaMA2 simply through initialization. We believe this constitutes a reasonable improvement.
>
> - Our additional results in Table 10 (Gemma-7B on GSM8K) show that NoRA outperforms LoRA by a larger margin (78.62 vs 76.72 in test accuracy).
>
> We remark that often even an improvement of 1\% is challenging; see e.g., [2].
> Given that initialization is, perhaps, the most lightweight modification to LoRA, it seems unfair to compare it with those heavily modified approaches. Moreover, *initialization is orthogonal to many existing methods, and thus holds great potential to be combined with other approaches for better performance.*
>
>
> **Q1.** What is the role of Gaussian playing in your analysis? To put it in other words, do you think it's possible to replace Gaussian with something, or Gaussian is essential for the analysis?
>
> **Response to Q1.** Thanks for pointing this out. The Gaussian $\mathbf{\Omega}$ is only for practical convenience. A random matrix from a Stiefel manifold can replace the Gaussian $\mathbf{\Omega}$. In fact, any $\mathbf{\Omega}$ that permits the conditions discussed a few lines before equation (3) would give the same quadratic rate. And the conditions are i) column of $\mathbf{X}_0$ is in the column space of $\mathbf{A}$, and ii) $\mathbf{X}_0$ is full rank. We have revised Sec. 2.2 accordingly.
>
>
> **References**
>
> [1] PEFT pacakge from HuggingFace. \url{https://huggingface.co/docs/peft/en/index}
>
> [2] https://huggingface.co/spaces/open-llm-leaderboard/open_llm_leaderboard

---

> > ### Comment · Reviewer_kgvD · 2024-11-20
> >
> > I thank authors for the response. Given that you don't actually need Gaussian for the analysis, I think it's quite important to emphasize and one could then attempt to improve the runtime of the algorithm by picking $\Omega$ to be a sparse embedding matrix [Nelson and Nguyen, FOCS'13] or a circulant transform matrix that could be applied to $A$ in sparsity of $A$ time, or $O(m^2 \log m)$ time. This potentially gives a nearly-linear time algorithm for matrix factorization under additive error scheme (note this could potentially be faster than SVD in theory).

---

> > > ### Author Response · Authors · 2024-11-21
> > > **Thank you for the insightful comment**
> > >
> > > We appreciate the reviewer for this very insightful comment. Indeed, the techniques proposed by the reviewer further reduce computational complexity.
> > >
> > > The reviewer's comment also inspires us. It turns out that the computational complexity, even in the dense regime, can be less than truncated SVD (which is ${\cal O}(n^2 r)$, here we assume $m=n$ for simplicity). We take asymmetric setting as an example.
> > >
> > > The key observation is that the computational burden of eq. (6) arises from matrix multiplication, whose complexity can be reduced with state-of-the-art algorithms [3, 4]. In particular, multiplication of two square matrix of size $n \times n$ can be done with complexity ${\cal O}(n^{2.38})$. To apply this on (6), we can pad the rectangular matrices with $0$ to make them square. Simple calculation shows that when $ r > {\cal O}(n^{0.38}) $, the complexity is lower than truncated SVD. Additionally, our derivation does not account for the sparsity of the padded matrices, suggesting that the theoretical computational complexity could be further reduced.
> > >
> > > We will explore the reviewer's comment to see whether it can further reduce the complexity. The manuscript will be updated accordingly.
> > >
> > > **Reference**
> > >
> > > [3] Vassilevska Williams, Virginia; Xu, Yinzhan; Xu, Zixuan; Zhou, Renfei. New Bounds for Matrix Multiplication: from Alpha to Omega. Proceedings of the 2024 Annual ACM-SIAM Symposium on Discrete Algorithms (SODA). pp. 3792–3835.
> > >
> > > [4] Alman, Josh; Duan, Ran; Williams, Virginia Vassilevska; Xu, Yinzhan; Xu, Zixuan; Zhou, Renfai (2024). "More Asymmetry Yields Faster Matrix Multiplication"

---

> > > > ### Comment · Reviewer_kgvD · 2024-11-21
> > > >
> > > > I think it's generally fair to assume $r<n^{\omega-2}$, as otherwise, you can just compute the full SVD in $O(n^\omega)$ time. On the other hand, when $r$ is very small ($r<n^\alpha$ for $\alpha\approx 0.31$), then multiplying $A$ with $\Omega$ takes $O(n^{2+o(1)})$ time. The interesting regime is where $n^{0.31} \leq r \leq n^{0.37}$, where computing $A\Omega$ takes super quadratic time if you pick $\Omega$ as Gaussian.

---

> > > > > ### Comment · Reviewer_kgvD · 2024-11-21
> > > > >
> > > > > Anyway, I think this is important to emphasize, as pointed by other reviewers, matrix factorization (without weights or binary mask as in matrix completion) seems a bit lack of motivations to study (though I agree studying it could bring many insights), as one can solve it directly using truncated SVD. But the computational advantage brought by using a structured $\Omega$ would make it an appealing alternative to truncated SVD.

---

### Official Review · Reviewer_jyg6 · 2024-11-01

**Soundness:** 3
**Presentation:** 3
**Contribution:** 3
**Rating:** 8
**Confidence:** 4

**Summary:**

The current work describes a novel initialization method for matrix factorization problems which accelerates convergence of ScaledGD both theoretically and empirically. The authors also consider extension of Nystrom initialization to LoRA model and shows its superiority compared to different baselines experimentally.

**Strengths:**

The current work studies how initialization affects ScaledGD method in matrix factorization problems, and considers its extension to LoRA model. I personally find the topic valuable. The theoretic bound improvement seems huge for classic matrix factorization problem. For LoRA model, recent work has studied customized optimizers and stepsize scheduler, NoRA proposes a new initialization scheme which has its theoretic grounds. Moreover, compared to PiSSA, NoRA is naturally zero-initialized and is thus a more convenient initialization to LoRA.

**Weaknesses:**

1. The convergence bound to under-parameterized case involves weak optimality. I'm not very familiar with this notion, the authors show all globally optimal solutions are weakly optimal, is there any intuition of this weak optimality or is it just curated from technical stuff? How larger is the space of weak optimality compared to true optimality?

2. Ideally to apply Nystrom initialization to LoRA, one should learn $\Delta W$, which suggests that one should know how much weight change is in order to initialize finetuning model, and is thus unrealistic.

**Questions:**

1. In Figure 2b, it kind of suggests that smaller $\xi$ is preferred for quicker convergence? So does it hint that Nystrom initialization benefits from small initializaion as well, which also induces zero saddle point concerns?

2. In NoRA, say we have two pretrained weights $W_0,W_1$ of pretty different magnitudes, does it indicate corresponding NoRA initialization also scales with such magnitudes since $X_\{0,1\}=W_\{0,1\}\Omega$? Probably adding certain weight normalization would make the finetuning more stable?


minor writing issues:
line 340: "modified ScaledGD in (6) have" should be "has"

---

> ### Author Response · Authors · 2024-11-18
>
> **W1.** is there any intuition of this weak optimality or is it just curated from technical stuff? How larger is the space of weak optimality compared to true optimality?
>
> **A1.** Weak optimality indeed arises for technical reasons. It means that the top-$r$ singular values are recovered subject to some bounded error related to other singular values. The intuition is that in the under-parametrization case, the smallest singular values would always perturb the convergence of the largest singular values. Theoretically, the weak optimal solution found by ScaledGD is at most ${\cal O}(r^{3/4})$ away from some global optimum; see more details in Lemma 5.
>
> Experimentally, our simulation in Fig. 1 (c) shows that there is indeed a regime (i.e., weak optimality) that ScaledGD slows down after reaching to it. For example, the purple curve have significantly different behaviors before and after 30 iterations.
>
>
> **W2.** Ideally to apply Nystrom initialization to LoRA, one should learn $\Delta \mathbf{W}$, which suggests that one should know how much weight change is in order to initialize finetuning model ...
>
> **A2.** For NoRA, although we do not know $\Delta \mathbf{W}$, a recent NeurIPS 2024 paper [Lingam et al. 2024] empirically shows that there exists a set of well-performed $\Delta \mathbf{W}$ whose columns space is subspace of the pretrained weight $\mathbf{W}_0$. Hence, we can employ $\mathbf{W}_0$ as an approximation for $\Delta \mathbf{W}$. We have polished Sec. 4 to make this clear.
>
> **Q1.** In Figure 1b, it kind of suggests that smaller $\xi$ is preferred for quicker convergence? So does it hint that Nystrom initialization benefits from small initializaion as well, which also induces zero saddle point concerns?
>
> **Response to Q1.** Thanks for pointing this out. The reason for different convergence speed arises from large $f(\mathbf{X}_0) -  f^*$. In other words, large $\xi$ with $\xi > 1$ coincidentally means large $f(\mathbf{X}_0) -  f^*$ for the numerical example in Fig. 1(b).
>
> We have updated Fig. 1(b) for avoiding confusion. Small initialization gives more complicated behaviors for saddle escaping, hence it does not necessarily lead to faster convergence; see the comparison between $\xi=1$ and the newly added red curve with $\xi=0.1$.
>
> **Q2.** In NoRA, say we have two pretrained weights of pretty different magnitudes, does it indicate corresponding NoRA initialization also scales with such magnitudes since? Probably adding certain weight normalization would make the finetuning more stable?
>
> **Response to Q2.** We thank the reviewer for this suggestion. We agree with this intuition that weight normalization could be helpful when the weight is significantly imbalanced. We are working on this for a systematic comparison.
>
> **Q3.** minor writing issues: line 340: "modified ScaledGD in (6) have" should be "has".
>
> **Response to Q3.** Thank you for pointing this out. We have corrected it and will double-check to ensure the manuscript is free of typos.

---

> > ### Author Response · Authors · 2024-11-24
> >
> > Dear reviewer, thank you once again for the time devoted to handling this work. Your insights have been helpful, and we sincerely appreciate your thoughtful review. Please don’t hesitate to let us know if there are any additional questions or points we can further clarify.

---

> > > ### Comment · Reviewer_jyg6 · 2024-11-27
> > >
> > > Thanks for the reply, I've increased my score.

---

### Official Review · Reviewer_FU9i · 2024-11-07

**Soundness:** 3
**Presentation:** 2
**Contribution:** 2
**Rating:** 5
**Confidence:** 5

**Summary:**

This paper studies the convergence of ScaledGD, a special preconditioned GD for matrix factorization (MF) problems, on symmetric/asymmetric MF problems. The authors show that initializing the factorized model as some random sketching of the target matrix significantly achieves faster convergence. Then, the authors propose to initialize the Low-rank adaptation (LoRA) using some random sketching of the pre-trained weights.

**Strengths:**

Good coverage on matrix factorization problems (symmetric/asymmetric, exact/over/under-parametrization); Writings are clear; Some connection to practical problems.

**Weaknesses:**

The reviewer has concerns regarding the significance and practical relevance of the theoretical results and the way results are connected to LoRA. Specifically:

1. The matrix factorization problem is not a practical problem to be solved, but rather a problem through which one builds a theoretical understanding of GD and its variants in a nonconvex setting. In this regard, any procedure using the target matrix itself, in the reviewer's opinion, is not allowed: if we know the target matrix, we can directly factorize it (which is why the MF problem is not a practical problem), thus any procedure (other than computing gradient, we are studying GD after all) using some information about the true target matrix is not allowed as it simply moves the factorized model one step closer to the target thus reduce the amount of work GD needs to converge. As such, the crucial role of initialization, as the authors stated, is just taking advantage of knowing the target matrix, which deviates from the original purpose of studying the MF problem: to understand how GD minimizes non-convex objectives even with random initialization. (Disclaimer: This point only pertains to MF problems. For practical problems such as matrix sensing, and matrix completion, the initializations can be special as they are non-trivially improving the convergence of GD)

2. as the reviewer already pointed out in 1., if we know the target matrix, then we can directly factorize it, which is exactly what the one-step modified ScaledGD with Nystrom initialization is doing, so Section 3 is trivial.

3. The connection to LoRA is puzzling in two ways: First, what is proposed is not remotely close to what is theoretically studied, Section 2, 3 studies initialization with the targe matrix sketched. The proposed initialization for LoRA is with the pre-trained weights sketched. Based on this, one should get the conclusion that "pre-trained weights" $\approx$ "targe matrix", which is clearly wrong. (Don't get too serious about this argument, the reviewer is just taking the shortcut to make the point). Moreover, the theoretical claims are special initialization accelerates GD, but none of the empirical experiments are about training time. On top of that, the performance improvement from special initialization seems very little to the reviewer.

**Questions:**

See "Weaknesses"

---

> ### Author Response · Authors · 2024-11-18
> **Responses to Reviewer FU9i (part 1/2)**
>
> **W1.** The matrix factorization problem is not a practical problem to be solved ... any procedure using the target matrix itself, in the reviewer's opinion, is not allowed ... the original purpose of studying the MF problem: to understand how GD minimizes non-convex objectives even with random initialization. Disclaimer: This point only pertains to MF problems ...
>
> **A1.** We thank the reviewer for the comment; however, we respectfully disagree with reviewer.
>
> 1. *Matrix factorization remains a legitimate problem to study*, and significant progress is still to be made. For example, the Lanczos method, which is commonly used in practice, achieves only a super-linear convergence rate under benign conditions. This is slower than the quadratic rate we achieve with our approach.
> Moreover, *matrix factorization is the population loss for matrix sensing, and is also closely related with matrix completion and tensor decomposition*. Our work provides new insights that can be possibly extended to these settings.
> Even though MF may not be directly practical, its theoretical analysis still provides valuable insights that are highly relevant for understanding optimization dynamics.
> We have updated Apdx. A.5 for further discussions.
>
> 2. *Our initialization is still valid even without direct access to $\mathbf{A}$*. It can be extracted from the gradients, where using $\mathbf{A}$ in gradient computation is the minimum requirement, as agreed by the reviewer.
> Taking the asymmetric setting as an example, the gradient of $\mathbf{X}$ is given by $(\mathbf{X} \mathbf{Y}^\top - \mathbf{A}) \mathbf{Y}$. It is straightforward that our initialization is the negative gradient of $\mathbf{X} = 0$ and $\mathbf{Y} = \bf{\Omega}$. This discussion is added in Apdx. A.4. Moreover, a recent NeurIPS publication [Ward and Kolda, 2023] also uses $\mathbf{A}$ for initialization. The result is well-received in the community.
>
> 3. It is important to note that $\mathbb{R}^{m \times n}$ is a large space, and  random small initialization is often used due to technical reasons [Du et al., 2018].  The study of optimization methods should not be restricted solely to small random initializations. The flexibility to use different initialization strategies, such as ours, could lead to better convergence behavior.
>
> 4. We believe many of our contributions are novel and have not appeared in prior literature. For example, initialization-dependent quadratic rate is not known, to the best of our knowledge. The quadratic rate of quasi-Newton methods (such as ScaledGD) is also not common even in smooth and strongly convex setting, and we have established it for a nonconvex and non-smooth problem. Under-parametrization is less studied in matrix factorization, while we give several results upon this topic.
>
> **W2.** as the reviewer already pointed out in 1., if we know the target matrix ...
>
> **A2.** We respectfully disagree with the reviewer for the same reasons stated in the previous response.
>
> We also hope to point out several design factors and implications that the reviewer possibly overlooked.
>
> 1. It is not clear from the first sight that whether one should initialize with $\mathbf{Y}_0=0$, or $\mathbf{Y}_0=1$.
>
> 2. The role of learning rate is also important. If a different one is chosen, ScaledGD does not share the same behavior; see Corollary 1.
>
> 3. Previous works often tackle the regime with $||\mathbf{X}_0 ||_F \approx || \mathbf{Y}_0||_F$, while our results provide new evidence that asymmetry helps asymmetric matrix factorization.

---

> > ### Author Response · Authors · 2024-11-18
> > **Responses to Reviewer FU9i (part 2/2)**
> >
> > **W3.** ... connection to LoRA is puzzling in two ways: First, what is proposed is not remotely close to what is theoretically studied, ... Moreover, the theoretical claims are special initialization accelerates GD, but none of the empirical experiments are about training time. On top of that, the performance improvement from special initialization seems very little to the reviewer.
> >
> > **A3.**
> > *Connection between the studied theory and LoRA setting.*
> >
> > - We provided further details on the links between our theories and its application to LoRA in Apdx. A.2. In particular, applying LoRA for least square problems with whitened data essentially is equivalent to the problem studied in Section 3. Note that whitened data is a common trick for theoretical convenience [Arora et al., 2018].
> >
> > - Recall that the key message from our theoretical results is that knowing the column space of the targeted $\Delta \mathbf{W}$ boosts convergence. Here we use the same notation as in (7). As we have discussed in Sec. 4, a recent NeurIPS 2024 paper [Lingam et al. 2024] empirically confirms that there is a set of well-performed $\Delta \mathbf{W}$ whose column space is subspace of the pretrained weight $\mathbf{W}_0$. Hence, $\mathbf{W}_0$ can be used as an estimation to $\Delta \mathbf{W}$. This aligns well with our message conveyed by theoretical findings. We have polished Sec. 4 to make this clearer.
> >
> >
> > *On iteration complexity.* The training time can be reflected via training loss, since per iteration time does not differ too much between LoRA and NoRA.
> > We do have reported the training loss in Table 3, where NoRA's training loss is reduced by roughly 10%.
> >
> > *Performance.* We hope to clarify our numerical results.
> >
> > - Sec. 5.1 shows that the proposed approach outperforms existing initialization approaches for LoRA. We note that the compared approaches, PiSSA and OLoRA have been integrated into HuggingFace's PEFT package [1], and are actively employed in the community.
> >
> > - Our results in Sec. 5.2 show that NoRA reduces the training loss of LoRA by 10\%. (This is an image generation task, hence we use loss to measure efficiency.)
> >
> > - For results in Sec. 5.3, note that LLaMA2 improves over LLaMA by 1.4 (in absolute accuracy). This is through significantly improved data quality and thousands of extra GPU hours. Our NoRA improves accuracy by 0.5 on LLaMA and 1.2 on LLaMA2 simply through initialization. We believe this constitutes a reasonable improvement.
> >
> > - Our additional results in Table 10 (Gemma-7B on GSM8K) show that NoRA outperforms LoRA by a larger margin (78.62 vs 76.72 in test accuracy).
> >
> > We remark that often even an improvement of 1\% is challenging; see e.g., [2]. Given that initialization is, perhaps, the most lightweight modification to LoRA, it seems unfair to compare it with those heavily modified approaches. Moreover, *initialization is orthogonal to many existing methods, and thus holds great potential to be combined with other approaches for better performance.*
> >
> > **References**
> >
> > [1] PEFT pacakge from HuggingFace. \url{https://huggingface.co/docs/peft/en/index}
> >
> > [2] https://huggingface.co/spaces/open-llm-leaderboard/open_llm_leaderboard

---

> > > ### Comment · Reviewer_FU9i · 2024-11-21
> > >
> > > Thanks for the response.
> > >
> > > * I acknowledge the author's comment, "Our initialization is still valid even without direct access to A. It can be extracted from the gradients." It should be a remark in the paper IMO that some validation with matrix completion/sensing with Nystrom initialization using estimated A would be good. I will raise the rating accordingly.
> > >
> > > * I disagree with the reviewer's response to Section 3. If what Section 3 is doing is just exactly factorizing A, just be open about it. Why state them as a gradient-based algorithm? Unless there are experiments showing the algorithm works for asymmetric matrix completion/sensing, where one can only estimate A.
> > >
> > > * I will leave the assessment of the experiments to other reviewers.
> > >
> > > Apologies for the late comment with some requests for experiments. I am open to revising the rating if the first two points are addressed.

---

> > > > ### Author Response · Authors · 2024-11-23
> > > >
> > > > We thank the reviewer for the feedback. And below please find our responses.
> > > >
> > > > **Factorization of gradient based method?**
> > > > If the learning rate $\eta=1$, indeed equation (6) can be understood as factorization. We see no conflict between factorization and gradient-based methods, as our work approaches the matrix factorization problem from an optimization perspective. In particular,
> > > >
> > > > - This factorization is derived from a gradient based approach, ScaledGD.
> > > >
> > > > - It is just convention in, e.g., optimization community. A similar example is that Newton's method is known to solve the least square problem $||Ax - b ||^2$ in a single step. Although coinciding with matrix inversion, it remains a valid Newton's iteration.
> > > >
> > > > **Extension to matrix sensing.** While this paper has not made any claims regarding matrix sensing, we address the reviewer’s request and discuss its extension to this setting.
> > > > For such problems where we only have access to measurements of the target matrix, our method can still be applied and at least achieves the same convergence rate as (Tong et al., 2021). This is because Nystrom initialization also satisfies Lemma 2 in (Tong et al., 2021).
> > > > Note that the proof in (Tong et al., 2021) for matrix sensing problems highly mirrors their proof for matrix factorization problems. Therefore, improvements in matrix factorization problems could also lead to better guarantees for matrix sensing problems. Since we provide the first known quadratic convergence rate for matrix factorization problems, it opens up the possibility of faster rates for matrix sensing problems as well. For example, a faster rate can be obtained for matrix sensing problems if we can estimate the column space of the target matrix from its measurements. We leave this for future investigation since the primary focus of the current work is matrix factorization.
> > > >
> > > > As suggested, we will include additional remarks in the updated manuscript. Please let us know if there are any further questions.

---

### Official Review · Reviewer_yocf · 2024-11-09

**Soundness:** 4
**Presentation:** 4
**Contribution:** 4
**Rating:** 5
**Confidence:** 3

**Summary:**

The paper shows that the low-rank matrix factorization problem can be significantly accelerated using a better initialization method. Specifically, the authors introduce the Nyström initialization, which uses a sketching idea to initialize the low-rank factors to be in the appropriate eigenspaces. Then, using Scaled Gradient Descent (ScaledGD), one can obtain quadratic convergence.

The authors then present Nyström-initialized LoRA (NoRA) and present experimental results that show the practical benefits of using such an initialization compared to using the standard LoRA.

**Strengths:**

On one hand, I feel that the quadratic convergence with the Nyström initialization is significant. I have not directly worked in the area of matrix completion, so I do not have full confidence in this judgment, but this type of quadratic convergence result seems quite important.

At the same time, however, I'm not quite sure if the quadratic convergence is coming from the scaled GD algorithm or from the Nyström initialization. Since the scaled GD algorithm is basically a Newton update, shouldn't the quadratic convergence happen asymptotically, regardless of the initialization? Is it that the Nyström initialization allows the algorithm to reach the quadratically convergent phase more quickly? Perhaps the authors can clarify this point.

**Weaknesses:**

On the other hand, I feel that the matrix completion result and the Nyström-initialized LoRA (NoRA) are strongly connected. The theory doesn't apply to the NoRA setting, but I do see that the Nyström initialization is a nice conceptual motivation for a potentially better LoRA initialization.

However, now with the flood of empirical LoRA papers, I think the community has a very high bar for recognizing work that purports to improve LoRA in a practical sense. I am not sure if the experimental validations of this work really clear that bar.

**Questions:**

.

---

> ### Author Response · Authors · 2024-11-18
>
> **Q1. the quadratic convergence with the Nystrom initialization is significant ... Since the scaled GD algorithm is basically a Newton update, shouldn't the quadratic convergence happen asymptotically, regardless of the initialization? Is it that the Nystrom initialization allows the algorithm to reach the quadratically convergent phase more quickly?**
>
> **A1.** The quadratic convergence is **determined by initialization**. Existing works of ScaledGD such as [Jia et al., 2023; Tong et al., 2021] only report a linear rate, and a quadratic rate is not observed in the experiments in these papers. It is the Nystrom initialization that leads to the quadratic rate of ScaledGD. This is corroborated in experiments in our Fig. 1(b), where it is observed that slightly perturbed Nystrom initialization slows down to linear convergence.
>
> We clarify that *ScaledGD is not a Newton method*, rather it can be viewed as a quasi-Newton approach, as discussed in [Tong et al., 2021]. To see this, notice that the Hessian of problems (1) and (4) is a tensor whose size does not match with the $r \times r$ precondition matrix $\mathbf{X}_t^\top\mathbf{X}_t$ used in ScaledGD. For quasi-Newton methods, quadratic rate is not known yet even in the strongly convex and smooth setting, and only non-asymptotic analyses for local rates are established recently; see e.g., [3, 4]. To provide further context, we have now added a paragraph  in Appendix A.1 to summarize the recent developments on quasi-Newton methods. This further highlights the significance of our results in achieving quadratic convergence for nonconvex and local-smooth problems.
>
>
> **Q2. ... the matrix completion result and NoRA are strongly connected. The theory doesn't apply to the NoRA setting, but I do see that the Nyström initialization is a nice conceptual motivation for a potentially better LoRA initialization. However, now with the flood of empirical LoRA papers, I think the community has a very high bar for recognizing work that purports to improve LoRA in a practical sense. I am not sure if the experimental validations of this work really clear that bar.**
>
> **A2.** Thanks for recognizing the strong connection between our matrix completion result and the NoRA setting. Our theories hold for applying LoRA on least squares, as discussed in Apdx A.2. Given the similarity in the bilinear structure between matrix factorization and LoRA, we consider matrix factorization a reasonable prototype problem for understanding the optimization dynamics of LoRA.
>
> **Performance.** We hope to clarify our numerical results.
>
> - Sec. 5.1 shows that the proposed approach outperforms existing initialization approaches for LoRA. We note that the compared approaches, PiSSA and OLoRA have been integrated into HuggingFace's PEFT package [1], and are actively employed in the community.
>
> - Our results in Sec. 5.2 show that NoRA reduces the training loss of LoRA by 10\%. (This is an image generation task, hence we use loss to measure efficiency.)
>
> - For results in Sec. 5.3, note that LLaMA2 improves over LLaMA by 1.4 (in absolute accuracy). This is through significantly improved data quality and thousands of extra GPU hours. Our NoRA improves accuracy by 0.5 on LLaMA and 1.2 on LLaMA2 simply through initialization. We believe this constitutes a reasonable improvement.
>
> - Our additional results in Table 10 (Gemma-7B on GSM8K) show that NoRA outperforms LoRA by a larger margin (78.62 vs 76.72 in test accuracy).
>
>
> We remark that often even an improvement of 1\% is challenging; see e.g., [2].
> Given that initialization is, perhaps, the most lightweight modification to LoRA, it seems unfair to compare it with those heavily modified approaches. Moreover, *initialization is orthogonal to many existing methods, and thus holds great potential to be combined with other approaches for better performance.*
>
> **References**
>
> [1] PEFT pacakge from HuggingFace. https://huggingface.co/docs/peft/en/index
>
> [2] https://huggingface.co/spaces/open-llm-leaderboard/open_llm_leaderboard
>
> [3] Rodomanov, Anton, and Yurii Nesterov. "Greedy quasi-Newton methods with explicit superlinear convergence." SIAM Journal on Optimization 31, 2021.
>
> [4] Jiang, Ruichen, Qiujiang Jin, and Aryan Mokhtari. "Online learning guided curvature approximation: A quasi-Newton method with global non-asymptotic superlinear convergence." In COLT, 2023.

---

> > ### Author Response · Authors · 2024-11-24
> >
> > Dear reviewer, thank you once again for the time devoted to handling this work. Your insights have been helpful, and we sincerely appreciate your thoughtful review. Please don’t hesitate to let us know if there are any additional questions or points we can further clarify.

---

> > ### Author Response · Authors · 2024-11-30
> >
> > As the rebuttal phase draws to a close, we would like to take this opportunity to sincerely thank the reviewer once again for handling our submission. We remain available and would be happy to address any further questions or concerns -- please let us know.

---

### Author Response · Authors · 2024-11-18
**Thank the reviewers for handling this submission, and also for the feedback**

We sincerely thank the reviewers for their time, effort, and constructive feedback on our manuscript. The comments and suggestions have been invaluable in enhancing both the clarity and scientific rigor of the paper. Below, we provide detailed point-to-point responses and clarifications, with the corresponding revisions highlighted in blue in the manuscript. We hope these revisions address your concerns. We remain open to any additional feedback or clarifications and look forward to an engaging and productive discussion.

---

### Meta-Review · Area_Chair_aj6L · 2024-12-18

**Metareview:**

This paper presents a Nyström initialization scheme for low-rank matrix factorization. The authors demonstrate that this initialization is critical for improving the convergence rates of the preconditioned ScaledGD algorithm in three different settings: (a) under-parameterized, (b) exactly-parameterized, and (c) over-parameterized. The paper addresses both symmetric and asymmetric matrices. Beyond deriving convergence rates for matrix factorization, the authors also explore applying the Nyström initialization in the LoRA setting.

The reviewers recognized the paper’s technical soundness, clarity of presentation, and overall contributions. During the rebuttal phase, the authors addressed most of the reviewers’ concerns, including the significance of the theoretical results in the non-symmetric case, the applicability of the results to more general problems (such as matrix sensing), and the relatively marginal improvements observed in the LoRA experiments. Although some issues remain and the paper could be further improved, I believe that (a) the overall quality of the work, (b) the insights into the importance of initialization for convergence rates, and (c) the contributions made are all clearly above the acceptance threshold. I therefore recommend its acceptance to ICLR.

**Additional Comments On Reviewer Discussion:**

During the rebuttal period, the authors responded to the reviewers’ concerns. These primarily focused on: (a) the significance of the theoretical contributions, especially in the asymmetric case (Section 3, Reviewer FU9i), (b) the generalization of the results beyond matrix factorization (e.g., to matrix sensing, Reviewer FU9i), and (c) the relatively marginal improvement observed in the LoRA experiments (Reviewers yocf and FU9i). The authors provided insights into how their approach could potentially be applied in more general settings, added a remark in the appendix, and further explained the non-trivial nature of the asymmetric results in Section 3. They also clarified why the experimental improvements in the LoRA setting offer practical benefits.

I agree with Reviewers yocf and FU9i that the paper could be improved by making the theoretical results more broadly applicable and providing more meticulous experimental evidence. However, I recognize that the insights presented in the paper are interesting (an opinion also shared by Reviewers kgvD and jyg6 (both scoring the paper an 8)] and have the potential to inspire further research into the role of initialization in accelerating convergence in various settings. Therefore, I recommend accepting the paper as a poster.

---

### Decision · Program_Chairs · 2025-01-22

Accept (Poster)